# Evaluating AI Grading on Real-World Handwritten College Mathematics: A Large-Scale Study Toward a Benchmark

Zhiqi Yu [1]   Xingping Liu [2]   Haobin Mao [3]   Mingshuo Liu [1]   Long Chen [1]   Jack Xin [1]   Yifeng Yu [1]

## Abstract

Grading in large undergraduate STEM courses often yields minimal feedback due to heavy instructional workloads. We present a large-scale empirical study of AI grading on real, handwritten single-variable calculus work from the University of California, Irvine. Using OCR-conditioned large language models with structured, rubric-guided prompting, our system produces scores and formative feedback for thousands of handwritten, in-person proctored free-response quiz submissions from nearly 800 students included in the paper's empirical analysis. In a setting with no single ground-truth label, we evaluate performance against official teaching-assistant grades, student surveys, and independent human review, finding strong alignment with TA scoring and a large majority of AI-generated feedback rated correct or acceptable across quizzes. Beyond calculus, this setting highlights core challenges in OCR-conditioned mathematical reasoning and partial-credit assessment. We analyze key failure modes, propose practical rubric- and prompt-design principles, and introduce a multi-perspective evaluation protocol for reliable real-course deployment. Building on the dataset and evaluation framework developed here, we outline a path toward a future standardized benchmark for AI grading of handwritten mathematics to support reproducible evaluation, transparent comparison, reliable deployment, and future research.

## 1. Introduction and Motivation

Grading at scale remains one of the most persistent challenges in undergraduate education. In high-enrollment STEM courses such as calculus, teaching assistants (TAs) face thousands of open-response submissions under tight deadlines. The resulting workload often leads to two systemic problems: (1) declining grading quality and consistency, and (2) minimal or no personalized feedback—although meaningful and informative feedback is essential for learning. These issues directly undermine fairness, transparency, and student engagement in foundational mathematics instruction. Since calculus is one of the largest and most resource-intensive gateway courses in STEM education nationwide, scalable grading solutions have potential impact across thousands of sections and tens of thousands of students each year.

To address this, we developed and deployed a large-scale AI grading system that combines optical character recognition (OCR) with large language models (LLMs) to assess handwritten solutions and generate rubric-aligned feedback. AI-generated scores were provided for reference only; TA scores remained the official course grades. To our knowledge, this is among the first large-scale real-course studies of LLM-based grading in university mathematics.

We deployed the system across three academic terms in a large-enrollment single-variable calculus sequence at UC Irvine, covering 20 in-person, proctored quizzes. The sequence consists of two consecutive courses– Math 2A and Math 2B.

1. **Winter 2025.** We tested the system on two quizzes in Math 2B, each containing two free-response questions, in a class of over 500 students.

2. **Spring 2025.** We applied the system to six quizzes in Math 2A (two questions each, one free-response and one multiple-choice or fill-in-the-blank, approximately 400 students) and six quizzes in Math 2B (two or three free-response questions, approximately 360 students). The Spring 2025 datasets served as the primary basis for system refinement; quizzes 2–6 from both Math 2A and Math 2B were used for detailed comparison

---

[1]Department of Mathematics, University of California, Irvine, Irvine, USA [2]Independent Researcher, Shanghai, China [3]IEOR Department, University of California, Berkeley, USA. Correspondence to: Yifeng Yu <yifengy@uci.edu>.

*Proceedings of the $43^{rd}$ International Conference on Machine Learning*, Seoul, South Korea. PMLR 306, 2026. Copyright 2026 by the author(s).

against TA scores, student surveys, and independent human reviewers.

3. **Fall 2025.** We further validated the system on six additional quizzes in Math 2B, each containing two or three free-response questions, with approximately 200 students.

All quantitative evaluations in this paper focus exclusively on free-response questions from 2025 Spring (more than 760 students), as multiple-choice and fill-in-the-blank items do not reflect the core challenges of AI-based handwriting interpretation and rubric-guided partial-credit assessment and the production of clear, constructive feedback.

We evaluate AI-assisted grading via TA-score alignment, student surveys, and an independent human review (over twenty reviewers). AI-assigned scores show strong agreement with human grading, and the generated feedback is generally judged correct or acceptable by independent reviewers, with remaining errors concentrated in OCR failures and a small set of rubric edge cases. Together, these results suggest that the system can reduce TA workload while providing more detailed feedback to students; see Section 6 for full quantitative results.

From a technical standpoint, one crucial aspect of grading handwritten mathematics with LLMs is prompt and rubric design. Because there is no unique ground-truth score for many partially correct solutions, the system must translate free-form handwritten student work into consistent, fine-grained judgments that remain aligned with course rubrics. Beyond coping with noisy OCR and varied handwriting, we therefore develop and iteratively refine a structured prompt framework that combines system-level instructions, problem-specific reference solutions, and machine-readable rubrics. This framework, described in Section 4, is crucial to stabilizing scores and generating useful feedback across thousands of heterogeneous submissions.

A growing body of recent work has studied AI-assisted grading of handwritten STEM solutions, especially with recent LLM advances, e.g., (Aisyah et al., 2025; Mok et al., 2024; Caraeni et al., 2025; Gandolfi, 2025; Lee et al., 2025; Liu et al., 2024; Nkoyo et al., 2025; Kortemeyer & Nöhl, 2025), but typically emphasizes score reliability rather than course-scale evaluation of feedback quality. We study course-scale deployment on thousands of authentic handwritten calculus submissions and evaluate both scores and feedback using TA grades, student surveys, and independent human review (20+ reviewers).

### Main Contributions

Our focus is on an underexplored ML problem: how to reliably evaluate and deploy LLM-based reasoning systems without ground truth, under noisy inputs, partial-credit grading, and constraints relevant to high-stakes deployment.

- **Large-scale, real-world deployment.** We report a large-scale deployment of an end-to-end OCR+LLM grading system (scores and feedback) on free-form handwritten calculus solutions in a high-enrollment, for-credit university course, using thousands of authentic quiz submissions.

- **Prompt-and-rubric grading pipeline.** We develop a practical pipeline that couples OCR-conditioned reasoning with carefully designed system prompts and a multi-rubric strategy (typically a flexible and a fixed rubric). The prompts and rubrics are iteratively refined using failure-case analysis to improve partial-credit scoring and formative feedback.

- **Multi-perspective human-in-the-loop evaluation.** We introduce a structured evaluation protocol combining TA score alignment, student feedback, and independent reviews from over twenty external reviewers to assess both score reliability and the accuracy/usefulness of AI-generated feedback, where no ground-truth labels exist.

- **Foundation for a future benchmark.** We have assembled a large collection of anonymized handwritten solutions together with OCR outputs, tuned rubrics, AI-generated scores and feedback, and independent human review results. As future work, we plan to release benchmark subsets in two tracks. *Track A (clean)* will pair scanned work with corrected transcriptions and instructor/TA–approved reference grades, enabling controlled evaluation of OCR and rubric-based grading. *Track B (noisy)* will retain the original OCR outputs and reviewer labels to evaluate robustness and human-review deferral—i.e., when the system should defer to human review under noisy inputs, and how reliably it can flag likely OCR-induced errors.

To clarify the scope of the present work, this paper delivers: (i) a large-scale real-course evaluation of an OCR+LLM grading system, (ii) empirical analysis of scoring, feedback quality, and failure modes, and (iii) a multi-perspective evaluation protocol for settings without ground-truth labels. The release of a public benchmark with standardized evaluation subsets, reference labels, and protocols is a future step, building on the dataset and protocol developed here.

Although this work focuses on single-variable calculus, the challenges it addresses—OCR robustness, grading reliability, scalability, and feedback quality—are shared across many large STEM courses, positioning large-enrollment entry-level mathematics as a natural testbed for broader AI-assisted assessment.

## 2. Overview of the AI Grading Pipeline

Our pipeline consists of three modular stages (Figure 1): region segmentation, OCR-to-LaTeX transcription, and rubric-based LLM grading with feedback. We use a standardized answer-sheet format that separates each problem into a solution region and a final-answer region, enabling the pipeline to handle variable numbers of problems by processing each region independently. See Appendix A.2. We perform OCR and grading at the region level (rather than on whole-page inputs) to reduce cross-problem interference and hallucination; free-form solution regions remain the most challenging. The resulting scores and feedback are delivered to students by email. See Appendix A.14 for an example.

**1. OCR and Transcription.** Handwritten student work is digitized via automated OCR and converted to LaTeX to preserve mathematical structure. Among the tools we evaluated, GPT-4.1-mini paired with well-designed prompts achieved a favorable trade-off among accuracy, scalability, and cost; see Section 3 for details.

**2. Rubric-Guided Prompting.** We use a two-level prompt design: a system message that provides global grading principles and output constraints, and problem-specific rubrics encoded as structured prompts specifying scoring and feedback criteria, output format, and exemplar solutions. Prompt templates are iteratively refined on sample submissions to better align LLM outputs with rubric intent (Section 4).

**3. LLM Evaluation.** The transcribed solutions and rubric prompts are processed by large language models, primarily GPT-4.1-mini, with supplementary evaluations using o3-mini if needed, which produce both numerical scores and explanatory feedback. Because grading lacks an absolute ground truth, we assess reliability via three complementary validations: alignment with TA scores, student feedback, and independent human reviews (Section 6).

**Model choice and evolving capabilities.** Unless otherwise noted, we use GPT-4.1-mini for both OCR and grading, balancing accuracy, latency, and cost for course-scale deployment. We additionally use o3-mini for a small subset of algebra-heavy or otherwise challenging cases where stronger reasoning is needed; see Section 5 for comparisons. Rather than optimizing for each model release, we keep the pipeline stable and allow the backend model to be swapped as capabilities improve. Systematic cross-model comparisons and routing policies are left to future work. We note that some OCR-related failure modes persist across newer models (see Appendix A.5).

## 3. OCR Component

For transcribing handwritten mathematical work, we evaluated several OCR options on our course data in Fall 2024/Winter 2025 and selected Mathpix as the most reliable choice at the time. After the release of GPT-4.1/4.1-mini in April 2025, we re-ran our OCR evaluation and found that, with OCR-specific prompting, GPT-4.1-mini produces higher-quality transcriptions on our data. Across the full set of handwritten submissions included in our empirical analysis, around 87.64% of transcriptions are *acceptable*, meaning they are either (i) *completely correct* (character-level LaTeX fidelity) or (ii) *sufficient for rubric-based grading*. Unlike Mathpix, which primarily targets literal transcription, GPT-4.1-mini can leverage contextual reasoning to resolve ambiguities, improving robustness to messy and heterogeneous student handwriting.

| OCR Method | Acceptable Transcriptions (%) |
|---|---|
| GPT-4.1-mini | 84 |
| Mathpix | 55 |

*Table 1.* OCR performance on the **171-solution subset**.

In a head-to-head comparison on a fixed challenging subset (the **171-solution subset**), consisting of 57 quizzes, each containing the same three questions (171 solutions in total), GPT-4.1-mini achieved 84% acceptable transcriptions, compared to 55% for Mathpix. See Table 2 for more comparisons and Appendix A.3 for representative examples.

Throughout this paper, the **171-solution subset** refers to this same fixed subset.

OCR performance with GPT-4.1-mini is sensitive to prompt design; we use context-aware constraints to improve transcription stability. Below is an example of our basic OCR prompt.

---
**OCR prompts**

This problem asks the student to estimate a definite integral using a left Riemann sum:

$$\int_1^5 (x^2 - x - 1)\, dx$$

with **four subintervals** and **left endpoints**, without evaluating the exact limit.

1. Use OCR to identify any text in this image and convert it into LaTeX code

2. Do not correct any math or logical error or typos.

---

To better utilize the contextual reasoning ability of GPT-4.1-mini OCR, it is important to state the problem in the prompts. The other two targeted prompt strategies address two common, prompt-addressable sources of transcription instability.

- **Hallucination.** When answer boxes are blank or sparsely written, GPT-4.1-mini may generate full solutions. The above Prompt 1 and pre-printing the

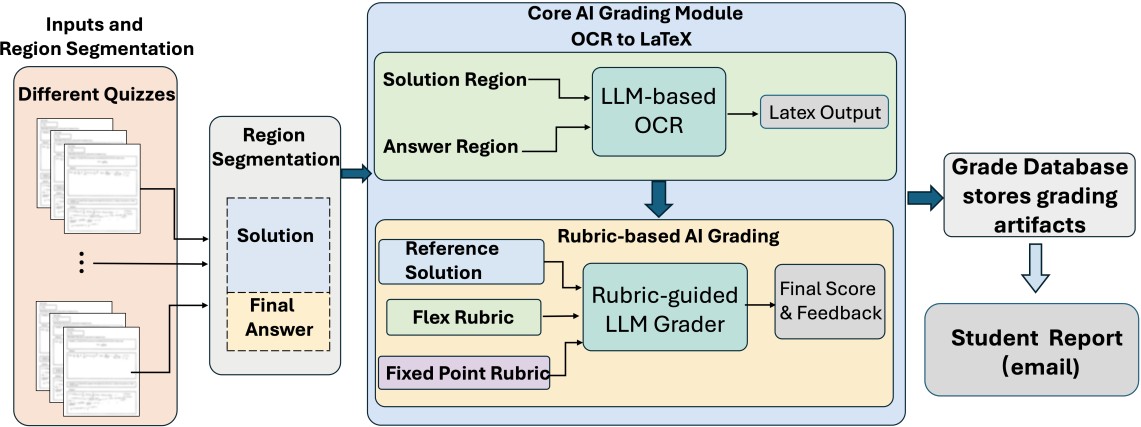

*Figure 1.* Overview of the implemented AI grading pipeline: The core module converts handwritten solutions to LaTeX and evaluates them against reference solutions and structured rubrics, producing detailed scores and feedback.

words "Solution" and "Final Answer" inside boxes stabilize the output and suppress hallucination (see Appendix A.2).

- **Autocorrection.** GPT-4.1-mini occasionally corrects students' mathematical mistakes during transcription. An explicit instruction not to correct errors (Prompt 2) helps reduce this behavior, though rare cases remain (less than 2% of our samples). A representative example is provided in Appendix A.5.

Two major OCR challenges remain: (1) geometric diagrams, where GPT-4.1-mini shows limited ability and, in our experiments, Mathpix did not reconstruct any diagrams from student work; and (2) reliably handling crossed-out or heavily scribbled text. In the latter case, OCR systems may transcribe erased work or drop nearby valid symbols when the crossed-out region visually merges with adjacent ink. Prompt refinements helped suppress erased content but sometimes also removed adjacent valid work.

Beyond these prompt-addressable issues, we analyzed all 27 problematic OCR cases within the **171-solution subset**. Most GPT-4.1-mini failures are layout-related, with the dominant failure mode involving complex division expressions. In particular, when nested fractions are written with insufficient spacing, the scope of the main fraction bar becomes ambiguous, causing terms to be merged, misparsed, or omitted. Other error types are uncommon. See Appendix A.4 for an illustrative example of this fraction-related failure mode.

To quantify the downstream impact of OCR quality on grading, we compare two pipelines on the 165 valid paired comparisons from the **171-solution subset**: Mathpix for OCR followed by GPT-4.1-mini for grading, and GPT-4.1-mini for both OCR and grading. The comparisons are conducted against two independent human reviewers as in Appendix A.10.3 and Appendix A.11. As shown in Table 2,

using GPT-4.1-mini OCR substantially reduces grading error: MAE decreases from approximately 0.69–0.71 with Mathpix to 0.40–0.48 with GPT-4.1-mini OCR, along with large gains in within-1 agreement. Since the grading model is held fixed, these results provide a controlled downstream comparison of OCR quality and demonstrate that OCR is a key bottleneck in end-to-end AI grading performance.

| Pair ($n = 165$) | MAE | Mean Diff | $\|d\| \leq 0.5$ (%) | $\|d\| \leq 1.0$ (%) |
|---|---|---|---|---|
| Mathpix+4.1-mini vs R1 | 0.712 | -0.239 | 60.61 | 78.79 |
| Mathpix+4.1-mini vs R2 | 0.691 | -0.630 | 61.21 | 76.97 |
| GPT-4.1-mini vs R1 | 0.482 | +0.094 | 76.36 | 90.30 |
| GPT-4.1-mini vs R2 | 0.400 | -0.297 | 80.61 | 90.91 |

*Table 2.* Impact of OCR quality on downstream grading. R1 and R2 refer to Reviewer 1 and Reviewer 2, respectively.

## 4. Grading and Structured Prompt Engineering

With OCR performance sufficiently reliable in our experiments, prompt design becomes central to consistent, fair, and rubric-aligned grading at scale. Careful system-message engineering and rubric formulation are key to achieving strong alignment with human grading.

### 4.1. System Message Design

Prompts in our grading system consist of two parts: a *system message* that defines grading behavior and a *rubric section* specifying problem-level criteria. We observed that API-based grading can be less thorough than grading performed in the interactive web interface. While the exact cause is not externally documented, this motivated us to make all grading instructions explicit in the system message so

that API-based deployment achieves consistent, high-quality outputs.

To reduce stochasticity, we set temperature to near-zero values (0.0–0.1) for GPT-4.1-mini to improve response stability and reproducibility. For o3-mini, temperature is not an exposed API parameter and therefore cannot be adjusted. The system message was iteratively refined to emphasize logical consistency, factual correctness, and human-style feedback. Below is an excerpt:

> ### System message excerpt
>
> *You are an expert math grader. When evaluating, please:*
>
> 1. *Distinguish between mathematically correct statements and incorrect reasoning.*
>
> 2. *Avoid contradictions in feedback.*
>
> 3. *Do not penalize false starts if later corrected.*
>
> 4. *Provide concise, constructive explanations for point deductions.*
>
> 5. *Write feedback in clear, natural language suitable for students.*

These structured instructions were designed to reduce logical inconsistency and excessive penalization of harmless earlier mistakes—common failure modes under naive prompting. A representative example of logical inconsistency is provided, together with further discussion of the prompt-design choices, in Appendix A.6.

### 4.2. Rubric and Prompt Framework

Our rubric framework, developed from thousands of Math 2A and Math 2B submissions, is designed to balance interpretability and grading precision. Each problem may be associated with multiple rubrics to accommodate different valid solution strategies. For most questions in this study, we adopt a dual-rubric design consisting of:

1. **Concise reference solutions**, summarizing one or more canonical correct approaches.

2. **Two complementary rubrics**:

   - **Flexible rubric** — human-like and tolerant of diverse reasoning styles; supports free-form evaluation to credit partially correct work and non-standard but valid approaches or presentations.

   - **Fixed rubric** — fine-grained and point-based, with problem-specific checks targeting required steps and common errors in standard solutions.

   See Appendix A.7 for a representative example comparing the two rubrics and their outputs, illustrating how

the flexible rubric complements a checklist-style rubric when student work is mathematically recoverable but not presented in a clean, standard form. Importantly, flexibility does not systematically inflate scores: in our experiments, when locally correct steps are globally inconsistent or numerically incorrect, the flexible rubric may assign lower scores than the fixed rubric.

3. **Selection rule.** To reduce model confusion, each submission is evaluated using both rubrics via independent API calls, and we adopt a *max-rule*: the higher of the two rubric scores is selected as the final grade, together with the corresponding feedback. We evaluate this design on Math 2A and Math 2B quizzes 5–6, which contain more challenging problems and admit a wider range of valid solution paths and partial attempts.

To assess alignment with human grading behavior, we compare grades produced by the fixed rubric alone, the flexible rubric alone, and the max-rule against TA scores (used as a reference rather than an absolute ground truth). Using mean absolute error (MAE) as the primary metric, the max-rule consistently achieves the lowest MAE across both courses and quizzes (Fig. 2). These results suggest that the max-rule yields more stable and human-aligned scores by crediting valid reasoning that may be under-recognized by a single rubric.

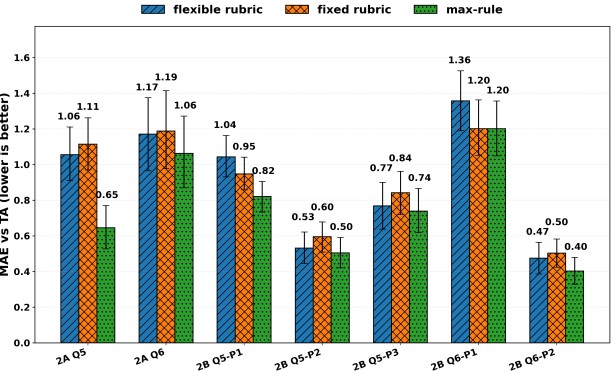

*Figure 2.* Bars correspond to the flexible rubric, fixed rubric and max-rule, respectively (left to right within each group).

**Exemplar-guided rubric generation.** We use examples to help the model write new grading rubrics in a consistent format. Using primarily Spring 2025 Math 2A and Math 2B quizzes, we built a small library of representative problems by problem type (e.g., definite integrals) and refined a high-quality rubric for each. For a new problem in the same or a closely related category in a later term (e.g., Fall 2025), we provide the problem statement, a reference solution, and the closest existing rubric to a large language model, and ask it to produce a rubric at the same level of detail. The resulting rubric typically needs only light human editing (e.g.,

adding a newly observed misconception or a course-specific convention) before deployment. Future work will organize these rubrics into a lightweight database and automate more of this rubric-writing step.

**Two sources of potentially unjustified point deductions.** We observed two recurring cases in which an LLM grader may deduct points despite correct underlying mathematics: penalizing disorganized or nonstandard presentation, and OCR errors in the final-answer box, which has less surrounding context than the solution area. In future deployments, we plan to add a human-review fallback when the solution clearly supports the correct answer but full credit is not assigned. In the reported experiments, our rubrics and system message already discourage style-based penalties and tolerate minor OCR-induced discrepancies in the final-answer box when the correct value is unambiguously supported by the solution steps; representative guidance is provided in Appendix A.6 and A.8.

## 5. o3-mini vs. GPT-4.1-mini Grading Variance and Targeted Accuracy Evaluation

We compare o3-mini and GPT-4.1-mini on grading accuracy and score variance using the algebra-heavy **171-solution set**. To isolate grader effects, we fix OCR to GPT-4.1-mini and vary only the grading model (GPT-4.1-mini vs. o3-mini). As o3-mini is a reasoning-oriented model, we expect it to perform better on algebraically involved items; two independent reviewers found feedback similarly accurate overall, but often preferred o3-mini when differences arose. However, o3-mini has higher run-to-run variability and cost, so we reserve it for more challenging items (Appendix A.10.3).

For such targeted cases, we use a simple **multi-run stabilization** procedure: when an item is re-evaluated by o3-mini, we run the model three times, compute the mean score, and retain the run whose score is closest to this mean, together with its feedback. This selection reduces run-to-run sensitivity while keeping the reported score and feedback from the same model run. The procedure is related in spirit to majority-vote self-consistency in LLM reasoning (Wang et al., 2023). See Appendix A.10.3 for further discussion and comparison.

**Human review.** If exactly one of the three runs assigns full credit while the other two do not, we flag the item for human review.

## 6. Evaluation Against Human Grading and Feedback

We evaluate AI-assisted grading for single-variable calculus in two courses (2A and 2B), covering nearly 800 students, and assess both score alignment and feedback quality. A central challenge is the absence of a ground-truth benchmark: there is no publicly available expert-graded dataset for calculus solutions, and human graders themselves show nontrivial disagreement, especially on borderline or unconventional answers (Appendix A.11). This variability makes it difficult to define a universally accepted "correct" score, and it makes feedback evaluation inherently subjective.

Given this inherent ambiguity, we adopt a pragmatic, multiperspective evaluation framework consisting of three complementary components:

- **Evaluating alignment with TA scores.** We compare AI-assigned scores with those assigned by teaching assistants (TAs). While TA grading can be inconsistent and often lacks detailed personalized feedback, it is the practical reference in university assessments, so alignment with TA practices is a necessary condition for deployment. Because the TA Gradescope rubric is designed for efficient human scoring, our AI rubric is adapted for automated interpretation and feedback; thus, we interpret score comparisons as measuring outcome alignment rather than strict rubric equivalence.

- **Direct student feedback.** We collect survey responses from Math 2A students on perceived correctness and usefulness. This helps identify potential misgrades and characterize how students interpret AI feedback, while recognizing that students may perceive the AI as overly strict when its scores are lower than the TA-assigned scores.

- **Independent review by external evaluators.** A team of trained graduate students, undergraduate students, and advanced high school students independently evaluates the AI outputs using the same rubrics applied by the AI, assessing both correctness and fairness and assigning their own scores. This provides an additional, partially decoupled, third-party perspective beyond TA alignment and student feedback.

Due to practical, technical, and administrative constraints (e.g., segmentation failures, scan quality issues, unmatched test codes, and reviewer availability), not all quizzes could be included in every comparison; these exclusions were determined prior to evaluation and are independent of grading outcomes (i.e., non-selective). See Appendix A.1 for a categorical breakdown.

Also, we evaluate agreement across all free-response submissions without stratifying by solution correctness, since OCR reliability and rubric-based reasoning must function even when the underlying mathematics is correct; grading alignment is therefore nontrivial in both correct and incorrect cases.

## 6.1. Alignment with TA Scores

We evaluate agreement between an AI grader and human TAs on 3,945 handwritten free-response student-question records from Math 2A and 2B. All Math 2B items are free response; in Math 2A, non-free-response items (e.g., multiple-choice or fill-in-the-blank items) are excluded from this analysis. Each record consists of an AI score, a TA score, and the gap $g = \text{AI} - \text{TA}$. We report the signed mean gap, $\mathbb{E}[g]$, and the mean absolute gap, $\mathbb{E}[|g|]$, also referred to as mean absolute error (MAE).

We summarize at two levels:

1. Global distribution of the score gap across all records;

2. Quiz-level aggregates: signed mean gap, MAE and percentage within 1 point ("within-1");

### 6.1.1. OVERALL AGREEMENT (GLOBAL)

The overall gap distribution is sharply peaked at 0, meaning AI and TAs give the same or very similar scores for many submissions (Figure 3). The histogram also has small tails: most differences are near 0, but a few submissions have large gaps. On average, AI scores are 0.40 points lower than TA scores, with a standard deviation of 1.12 (about one point).

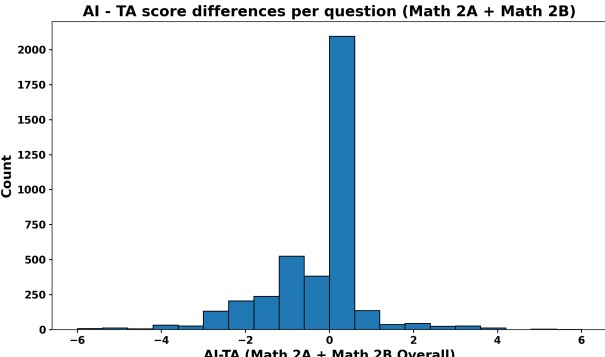

**AI - TA score differences per question (Math 2A + Math 2B)**

*Figure 3.* Global distribution of AI–TA score gaps.

### 6.1.2. QUIZ-LEVEL SUMMARY

Across quizzes, the mean absolute gap (MAE) ranges from 0.47 to 1.06 points, and the within-1 rate ranges from 68% to 86%, demonstrating strong overall alignment with TA grading. See Table 3.

We additionally report 95% confidence intervals (CIs) for quiz-level MAE estimates in Appendix A.12, computed using 5000 bootstrap resamples. The intervals are generally narrow, particularly for quizzes with larger sample sizes, indicating that the MAE estimates are statistically stable under resampling and not highly sensitive to sampling variability.

| Course | Quiz | # Q | # Rec | $\mathbb{E}[g]$ | MAE | Within 1 (%) |
|---|---|---|---|---|---|---|
| 2A | Quiz 2 | 2 | 540 | -0.47 | 0.72 | 82% |
| 2A | Quiz 3 | 1 | 265 | 0.23 | 0.71 | 82% |
| 2A | Quiz 4 | 2 | 308 | -0.23 | 0.78 | 77% |
| 2A | Quiz 5 | 1 | 255 | -0.38 | 0.65 | 83% |
| 2A | Quiz 6 | 1 | 88 | -0.05 | 1.06 | 68% |
| 2B | Quiz 2 | 3 | 402 | -0.48 | 0.71 | 81% |
| 2B | Quiz 3 | 3 | 618 | -0.39 | 0.47 | 86% |
| 2B | Quiz 4 | 3 | 453 | -0.71 | 0.83 | 72% |
| 2B | Quiz 5 | 3 | 624 | -0.41 | 0.69 | 78% |
| 2B | Quiz 6 | 2 | 392 | -0.59 | 0.80 | 77% |

*Table 3.* Quiz-level summary of AI–TA agreement. Here # Q and # Rec are numbers of questions and records, respectively, and within 1 (%) is the percentage of records with $|g| \leq 1$.

For larger discrepancies with $|g| > 1$, negative gaps ($g < -1$) are primarily attributable to OCR errors or occasional leniency in TA grading (i.e., the AI is comparatively harsher). See Appendix A.9 for a representative positive outlier $g > 1$ and further discussion.

To assess whether the max-rule aggregation introduces systematic bias, we examine the signed mean gap ($g = \text{AI} - \text{TA}$) across TA–quiz sections (25 sections in Math 2A and 15 sections in Math 2B). The differences are predominantly negative, indicating that AI grading is slightly stricter than TA grading. This pattern is consistent across sections and is not driven by any single TA or quiz (see Appendix A.13 for details).

## 6.2. Survey Results from Students

The following summarizes results from a voluntary student survey regarding the feedback provided by our grading system. The survey was conducted in two sections of Math 2A (total 301 respondents out of approximately 400 enrolled students). No survey was administered in Math 2B. Interestingly, the two independent sections exhibit highly consistent response patterns across all items.

| Math 2A | Accurate | Clear | Prefer TA | Open to AI |
|---|---|---|---|---|
| Section 1 (n=158) | 60% | 63% | 52% | 78% |
| Section 2 (n=143) | 61% | 62% | 53% | 76% |

*Table 4.* Student survey summary. "Accurate" = feedback very or mostly accurate; "Clear" = feedback very or mostly clear; "Open to AI" = yes or maybe with improvements.

Overall, the survey suggests cautious openness toward AI-assisted grading. Open-ended responses raised concerns about handwriting recognition (OCR), partial-credit strictness, and limited explanations, and emphasized the need for human oversight. These comments motivate robust OCR and an automatic review-triggering mechanism that escalates low-confidence cases for manual checking (Section 7).

## 6.3. Agreement Between AI and Independent Human Reviewers

While AI–TA score agreement was already high, score matching alone does not guarantee correctness or feedback quality. We therefore employed independent reviewers to review the AI outputs on 3,851 handwritten free-response student–question records (Math 2A: 1,819; Math 2B: 2,032) by verifying (i) input quality (OCR legibility), (ii) score alignment under the same grading rubric, and (iii) rubric fidelity and feedback accuracy. Reviewers evaluated the AI outputs without access to the corresponding TA scores. This provides an independent, rubric-consistent evaluation beyond raw TA alignment, which is valuable given the lack of an objective ground-truth score for handwritten free-response work.

**Input Quality (OCR).** Most scripts were readable: 87.64% were rated *Acceptable* and 12.36% *Problematic* (Figure 4). See Section 3 for typical failure modes.

**Score Agreement.** Across 3,851 graded responses, the score difference (AI–Reviewer) is sharply centered at 0 with relatively light tails on both sides; over 90% lie within $\pm 1$ point (Figure 5), indicating strong numerical concordance.

**Rubric Fidelity and Feedback Accuracy.** Independent review found 79.79% of AI evaluations fully correct, 9.55% acceptable, and 10.67% incorrect (Figure 6). Beyond OCR-related failures, two main error patterns emerged: (i) over-penalizing valid solution methods that are mathematically correct but not explicitly listed in the rubric, and (ii) over-crediting rubric-shaped explanations that follow familiar steps but contain incorrect reasoning. This latter pattern is also familiar in human grading under time constraints, where rubric-shaped work may receive credit despite subtle reasoning errors.

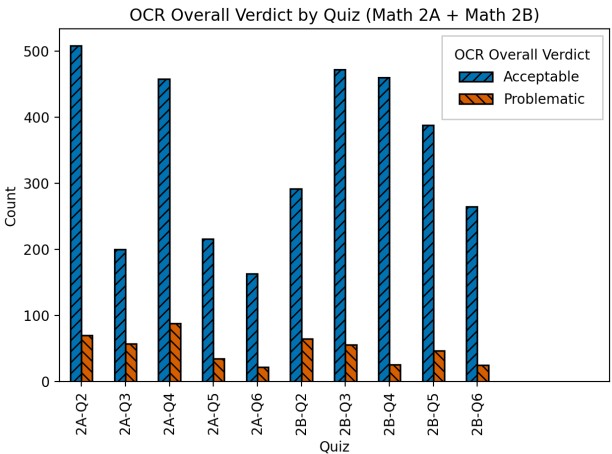

*Figure 4.* OCR verdict distribution.

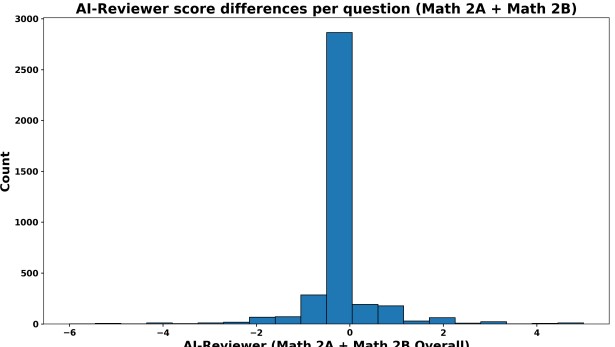

*Figure 5.* Global distribution of AI-Reviewer score gaps.

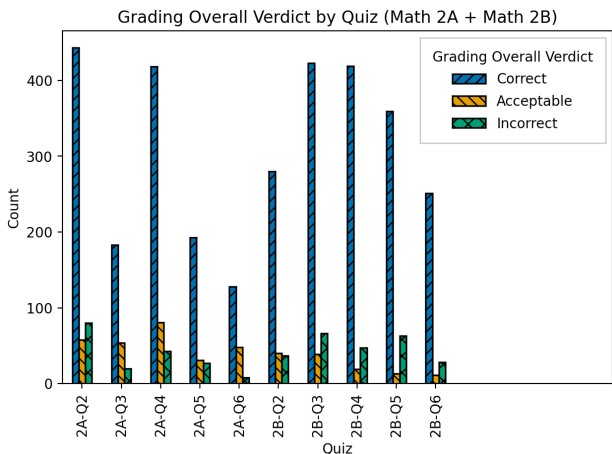

*Figure 6.* Rubric fidelity and feedback review distribution.

Using the same two independent human reviewers as in Appendix A.10.3 and A.11, we compare AI–human agreement with human–human agreement on the **171-solution subset** (Table 5). The AI–human MAE is comparable to the inter-reviewer MAE, and the within-1 agreement rates are also similar or higher. This suggests that, on this subset, AI grading is within the range of observed human–human variability. See Appendix A.11 for additional information on intraclass correlation coefficient (ICC) and $p$-values.

| Pair ($n = 171$) | MAE | Mean Diff | $|d| \leq 0.5$ (%) | $|d| \leq 1.0$ (%) |
|---|---|---|---|---|
| 4.1-mini vs R1 | 0.485 | +0.094 | 76.02 | 90.64 |
| 4.1-mini vs R2 | 0.401 | -0.301 | 80.12 | 91.23 |
| o3-mini vs R1 | 0.456 | +0.211 | 73.10 | 93.57 |
| o3-mini vs R2 | 0.237 | -0.184 | 92.40 | 97.66 |
| R1 vs R2 | 0.477 | -0.395 | 75.44 | 90.64 |

*Table 5.* Agreement statistics between human reviewers and AI graders. R1 and R2 denote Reviewer 1 and Reviewer 2, respectively. For each row $X$ vs $Y$, $d = X - Y$. The columns $|d| \leq 0.5$ and $|d| \leq 1.0$ report the percentage of cases within the corresponding thresholds.

# 7. Limitations

Our study has several limitations that point to clear next steps.

**Intervention policy.** A key bottleneck for large-scale deployment is deciding when to trust automated grading versus defer to additional checks or human review. While OCR errors and rubric–prompt interactions remain major sources of failure, we do not yet have a systematic method for deciding when to trigger such checks or defer to human review. In our experiments, we flag only a narrow set of issues (e.g., a correct final answer receiving less than full credit, or high variance across repeated runs), and these triggers are simple, manually designed heuristics rather than a unified decision rule. Related ML work on learned deferral, uncertainty estimates for safer downstream decisions, and fair abstention offers principled perspectives for triggering human review, e.g., (Madras et al., 2018; Kiyani et al., 2025; Casacuberta & Kanade, 2025), but adapting these frameworks to OCR-conditioned, partial-credit grading without ground truth remains open.

**High-stakes deployment.** Our experiments focus on low-stakes quizzes ($\approx$10% of the course grade); extending AI-assisted grading to midterms and finals would raise the cost of errors and require substantially more conservative human-in-the-loop controls.

**Choosing the most accurate grade/feedback.** When multiple OCR engines or graders are available (e.g., GPT-based OCR vs. Mathpix, or different LLM graders), we currently rely on human reviewers to resolve disagreements and decide which output is more accurate. This does not scale and also slows our ability to evaluate or swap in new models. Developing a reliable automatic mechanism to detect disagreements and select the most reliable score+feedback is a future project.

**Rubric aggregation (max-rule).** We aggregate the fixed and flexible rubric scores using the max-rule. Among the tested rubric scores (fixed, flexible, and max-rule), this choice achieves the lowest MAE against TA grades. Although selecting the higher score could inflate scores in principle, we do not observe systematic inflation: max-rule scores remain lower than TA scores on average. Since convex combinations of the two rubric scores cannot exceed the max-rule score, they would tend to lower scores further in our setting. We leave broader comparisons with other aggregation rules to future work.

**Scope.** Our evaluation focuses on single-variable calculus at one institution; additional deployments across courses and institutions are needed to establish generalizability and to understand operational impact under different grading cultures and student populations.

**Fairness and strategic adaptation.** Although our data come from proctored, in-person assessments, fairness-related risks remain: OCR errors may disproportionately affect students with messier handwriting, nonstandard notation, or unconventional explanations. We have not yet conducted systematic subgroup-level fairness analysis, partly due to the lack of demographic labels and the difficulty of quantifying handwriting quality. A related concern is strategic adaptation, where students may learn to present solutions in rubric- or model-preferred forms. Addressing these risks will require OCR-aware deferral rules, targeted fairness review, and checks against rubric-shaped but incorrect reasoning, including possible prompt-injection attempts or other forms of strategic adaptation.

# 8. Conclusion

We presented a large-scale evaluation of AI-assisted grading for single-variable calculus at UC Irvine, covering thousands of quiz solutions from nearly 800 students. Our system integrates OCR, prompt-guided reasoning, and rubric-based scoring in an automated workflow, and was evaluated through comparisons with TA grades, structured human review, and student feedback.

Our results suggest that current AI systems can already serve as reliable grading assistants: they handle most routine problems accurately, provide usable feedback, and offer a practical path toward reducing manual grading effort under appropriate human oversight. Key challenges remain, including OCR failures on diagram- or geometry-heavy work and cancellations, consistent partial-credit assignment across diverse solution variants, and principled policies for triggering human review in low-confidence cases.

Looking ahead, we plan to release well-documented benchmark subsets from the database collected in this study to support reproducible evaluation of OCR and rubric-based grading. Specifically, we will provide two tracks: **Track A** (clean) with corrected transcriptions and instructor/TA–approved reference grades, and **Track B** (noisy) retaining original OCR outputs and reviewer labels to assess robustness and inform human-review decisions (i.e., when to defer to human review). We hope these resources will enable more transparent and scalable comparisons, support the evaluation and deployment of AI graders, and accelerate progress toward reliable automated assessment in large-enrollment STEM courses.

## Acknowledgments

We gratefully acknowledge the UC Irvine School of Physical Sciences and the Department of Mathematics for their continued support of this AI grading project. We are grateful to the anonymous reviewers for their careful reading and constructive feedback, which helped improve the presentation, analysis, and positioning of this work. We are deeply grateful to our graduate student reviewers—Jongwon D. Kim, Yuanduan Lou, Shiyi Luo, and Zeyi Xu; our undergraduate student reviewers—Keith Bui, Xin Gao, Changze Li, Tony Li, Alex Lu, Eric Lyu, Xuaner Niu, Oleg Presnyakov, Scott Sibley, Vincent Wang, Qian Zhang, and Mickey Zhao; and our high school student reviewers—Eric Guo, Kevin Huang, Aaron Liu, Jasper Sun, An My Ngoc Tran, and Lauren Zhang—for reviewing AI grading results and providing constructive feedback and suggestions. We especially thank Oleg Presnyakov for suggesting the max-rule (selecting the maximum score across multiple rubrics). Special thanks are due to the MATH 2A/2B instructors—Rachel Cohen Lehman, Ziming Shi, Daniel V. Simonson, and Yang Zhang—for their generous support, and to the MATH 2A/2B teaching assistants for their valuable assistance throughout this project. Long Chen was partially supported by NSF grant DMS-2309777, Jack Xin was partially supported by NSF grant DMS-2309520, and Yifeng Yu was partially supported by NSF grant DMS-2000191.

## Impact Statement

This paper presents work whose goal is to advance the field of machine learning through the study of OCR-conditioned large language models for rubric-guided grading of handwritten mathematics in real university courses. The primary potential societal benefit is improved scalability, consistency, and feedback quality in large-enrollment STEM education, which may reduce instructional workload and help provide more timely and informative feedback to students. At the same time, automated grading systems raise ethical considerations related to fairness, robustness, and accountability, particularly when errors arise from OCR failures, non-canonical solution styles, or ambiguous partial-credit judgments. To mitigate these risks, the system studied here was used only as a decision-support tool, with human instructors retaining full control over official grades, and was evaluated using multiple complementary human-centered validation protocols. We emphasize that responsible deployment requires continued human oversight, conservative deferral policies, transparency to students, and careful attention to bias and error modes, especially in higher-stakes assessments.

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

# A. Appendix

## A.1. Dataset Coverage Across Validation Protocol

Due to practical technical and administrative constraints, not all submitted quizzes could be included in every validation protocol. We distinguish (i) *universal* exclusions that apply to any downstream analysis, and (ii) additional constraints specific to particular protocols (e.g., AI–TA score comparison or independent human review). Exclusions were driven by processing or logistical feasibility rather than by grading outcomes, and are intended to be non-selective with respect to score disagreement.

**Universal exclusions (apply to all protocols).** First, some quizzes were not successfully processed by our pipeline (most commonly due to segmentation failures when students wrote outside designated boxes, scan-quality issues, or occasional OCR/input-extraction failures). In addition, as part of model iteration and coordination with course staff, the first quiz in both Math 2A and Math 2B was excluded from our evaluation sets. Finally, one Math 2B TA section exhibited systematic scanning artifacts (e.g., extraneous lines on scanned pages) that materially degraded segmentation and input extraction; we therefore excluded that section from all analyses.

**Additional constraints for AI–TA score comparison (record linkage).** The AI–TA comparison further requires linking each processed submission to a Gradescope record via the assigned de-identification test code. This linkage sometimes fails because a subset of students omitted or mistyped the code, and in some cases TA score exports from Gradescope were incomplete or missing for reasons outside our control, preventing a reliable per-question score comparison.

**Additional constraints for independent human review (review logistics).** The independent review was organized at the TA-section level: each reviewer reviewed quizzes from a single TA section, and not all TA sections could be covered; for example, when a TA taught multiple discussion sections, we sometimes reviewed only a subset of those sections due to limited reviewer capacity. A small fraction of assigned reviews were not completed because some reviewers did not finish them.

## A.2. Design of Answer Sheets

Below are two answer-sheet designs. Version 1 allows blank answers, which can trigger hallucinations despite explicit OCR instructions. Version 2 reduces this by placing "**Solution**" and "**Final Answer**" in designated boxes.

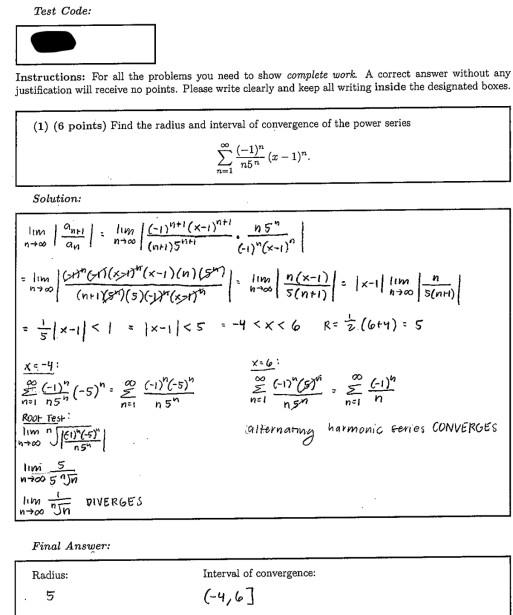

*Figure 7.* Answer Sheet Version 1

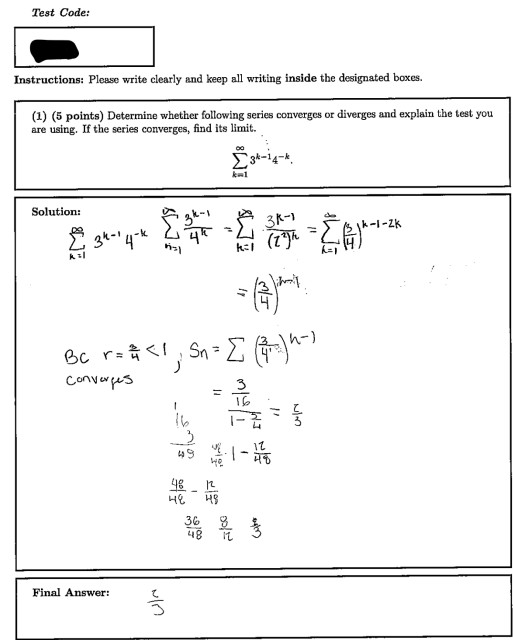

*Figure 8.* Answer Sheet Version 2

## A.3. GPT-4.1-mini vs. Mathpix

In what follows, we present several representative examples showing that (1) GPT-4.1-mini demonstrates strong performance, (2) illustrative comparisons between Mathpix and GPT-4.1-mini outputs highlight key differences, and (3) how to effectively mitigate issues such as autocorrection.

**Example 1.** Read the following handwritten solution from a Math 2B quiz. The problem is to find the sum of the series:

$$\sum_{n=3}^{\infty} \frac{2^{n-1}}{3^{n+3}}.$$

**OCR output from GPT-4.1-mini, returned as LaTeX via our API pipeline and rendered here as a compiled PDF.** The transcription is nearly perfect and the correct score is assigned by the downstream evaluation.

$$\frac{2^{n-1}}{3^{n+3}} = \frac{2^n \cdot 2^{-1}}{3^n \cdot 3^3} = \frac{1}{2 \cdot 27} \times \left(\frac{2}{3}\right)^n = \frac{1}{54} \cdot \left(\frac{2}{3}\right)^n$$

$$\sum_{n=3}^{\infty} \left(\frac{1}{54} \cdot \left(\frac{2}{3}\right)^n\right) = \frac{1}{54} \sum_{n=3}^{\infty} \left(\frac{2}{3}\right)^n \quad \Rightarrow \textit{first term } a = \left(\frac{2}{3}\right)^3 = \frac{8}{27}$$

*ratio:* $r = \frac{2}{3}$ & $|r| < 1$

$$\sum_{n=3}^{\infty} \left(\frac{2}{3}\right)^n = \frac{\frac{8}{27}}{1 - \frac{2}{3}} = \frac{8}{27} \times 3 = \frac{24}{27} = \frac{8}{9} \quad \Rightarrow \textit{it converges}$$

$$\frac{1}{54} \cdot \frac{8}{9} = \frac{8}{486} = \frac{4}{243}$$

This example demonstrates the reasoning-based reading capability of GPT-4.1-mini. From the surrounding context, the model correctly inferred the student's intended expressions: (1) it repaired a minor notational omission by inserting missing parentheses, interpreting the term as $\left(\frac{2}{3}\right)^n$; and (2) it resolved an ambiguous notation in the denominator (the minus sign is faint or possibly absent) of $\frac{\frac{8}{27}}{1-\frac{2}{3}}$ rather than interpreting it as the mixed number $1\frac{2}{3}$. We therefore classify the output as *almost perfect*, reflecting correct intent recovery rather than literal transcription. In this case, a strictly literal reading could lead to unwarranted point loss during grading.

**OCR output from Mathpix, returned as LaTeX via our API pipeline and rendered here as a compiled PDF:** *The transcription contains several typos, which leads to unwanted one point loss in the downstream evaluation part.*

$\frac{2^{n-1}}{3^{n+3}} = \frac{2^n \cdot 2^{-1}}{3^n \cdot 3^3} = \frac{1}{2 \cdot 27} \times \left(\frac{2}{3}\right)^n = \frac{1}{54} \cdot \frac{2}{3}n$

$\sum_{n=3}^{\infty} \left(\frac{1}{54} \cdot \left(\frac{2}{3}\right)^n\right) = \frac{1}{54} \sum_{n=3}^{\infty} \left(\frac{2}{3}\right)^n \Rightarrow \textit{firsterm } a = \left(\frac{2}{3}\right)^3 = \frac{8}{27}$

$\frac{k}{\infty} \left(\frac{2}{3}\right)^n = \frac{8}{27} = \frac{8}{27} \times 3 = \frac{24}{27} = \frac{8}{9} \quad \Rightarrow \textit{it converges}$

$n = 3 \quad \frac{8}{34} \cdot \frac{8}{9} = \frac{8}{486} \cdot \frac{4}{243}$

$\frac{1}{54}$

Out of curiosity, we also processed the above sample on Mathpix Snip, the online version. Below is one output (rendered in PDF).

$$\frac{2^{n-1}}{3^{n+3}} = \frac{2^n \cdot 2^{-1}}{3^n \cdot 3^3} = \frac{1}{2 \cdot 27} \times \left(\frac{2}{3}\right)^n = \frac{1}{54} \cdot \frac{2}{3}$$

$$\sum_{n=3}^{\infty}\left(\frac{1}{54}\cdot\left(\frac{2}{3}\right)^n\right) = \frac{1}{54}\sum_{n=3}^{\infty}\left(\frac{2}{3}\right)^n \Rightarrow \text{first term } a = \left(\frac{2}{3}\right)^3 = \frac{8}{27}$$

$$\frac{k}{n}\left(\frac{2}{3}\right)^n = \frac{\frac{8}{27}}{1\frac{2}{3}} = \frac{8}{27} \times 3 = \frac{24}{27} = \frac{8}{9} \quad \Rightarrow \text{ it converges}$$

$$\frac{1}{54} \cdot \frac{8}{9} = \frac{8}{486} \therefore \frac{4}{243}$$

**Example 2.** Below is another example demonstrating the strength of GPT-4.1-mini in reading messy handwritten English words. The problem is: **(4 points)** Determine whether the series

$$\sum_{n=1}^{\infty} \frac{(-1)^n}{\sqrt{n}}$$

converges or diverges. Show all work and state the test you are using.

Note that the handwritten English words are not easy to recognize, even to human readers.

**OCR output from GPT-4.1-mini, returned as LaTeX via our API pipeline and rendered here as a compiled PDF:** *The transcription is almost perfect: it is perfect in the solution part, with a single typo in the final answer (likely due to the lack of context).*

*Solution Part:*

$$\sum_{n=1}^{\infty} \frac{(-1)^n}{\sqrt{n}} \quad a_n = \frac{1}{\sqrt{n}} \implies a_n \frac{1}{n^{1/2}}$$

*p-series $\frac{1}{2} < 1$ Divergent*

*Due to the p-series $\leq 1$, due to the Alternating Series Rule, $\sum_{n=1}^{\infty} \frac{(-1)^n}{\sqrt{n}}$ must also be divergent.*

*Final answer: Dirugnt.*

Although the final answer contains a typo, the downstream evaluation module still correctly interprets "Dirugnt" as a misspelling of "divergent".

**OCR output from Mathpix, returned as LaTeX via our API pipeline and rendered here as a compiled PDF:** In contrast, the Mathpix transcription contains many typos in both mathematical notation and English words:

*Solution part: $\sum_{n=1}^{\infty} \frac{(-1)^n}{\sqrt{n}}$     an $a_n = \frac{1}{\sqrt{n}} \Rightarrow a_n \frac{1}{n^{1/2}}$ penns $1/2 < 1$ Divergent*

*B/c Dare to the p-serves $< 1$, duc to the Alternahy sums thate, $\sum_{n=1}^{\infty} \frac{(-1)^n}{\sqrt{n}}$ mist also be dougunt.*

*Final answer part: Direngut.*

We would like to remark that despite the numerous word-level typos in Mathpix output, the downstream model is still able to recover the student's intended reasoning from context. Nevertheless, there are examples where confusion caused by misspelling did contribute unwanted point loss.

**Example 3.** This example shows that GPT-4.1-mini can correctly read very faint handwriting and produce a correct downstream interpretation, whereas Mathpix fails to do so, leading to unwarranted point loss in the downstream evaluation.

**OCR output from GPT-4.1-mini, returned as LaTeX via our API pipeline and rendered here as a compiled PDF.:** The transcription is perfect.

**Solution Part:**

$$b_n = \frac{1}{\sqrt{n}} \qquad -\frac{1}{1}, \frac{1}{\sqrt{2}}, -\frac{1}{\sqrt{3}}, \frac{1}{\sqrt{4}}$$

$$b_n \leq b_{n-1}\checkmark$$

$$\lim_{n\to\infty} b_n = 0\checkmark$$

*alternating series test $\Rightarrow$ shows it is convergent by satisfying conditions.*

**Final answer Part:** *Convergent*

**OCR output from Mathpix, returned as LaTeX via our API pipeline and rendered here as a compiled PDF***:* The transcription contains multiple typos in both mathematical notation and English words, and it misses a key part of the test name, which results in a 1-point loss (out of 4) in the downstream evaluation.

**Solution Part:**

$$b_n = \frac{1}{\sqrt{n}} \quad - \frac{1}{1}, \frac{1}{\sqrt{2}}, -\frac{1}{\sqrt{3}}, \frac{1}{\sqrt{4}}$$
$$b_n \leq b_{n-1}\sqrt{}$$
$$\lim_{n\to\infty} \to b_n = 0/$$

*curcitiums*

**Final Answer Part:** *Convergent*

## A.4. Failure Example of GPT-4.1-mini OCR

The problem is **(3 points)** Find the sum of the series

$$\sum_{n=3}^{\infty} \frac{2^{n-1}}{3^{n+3}}.$$

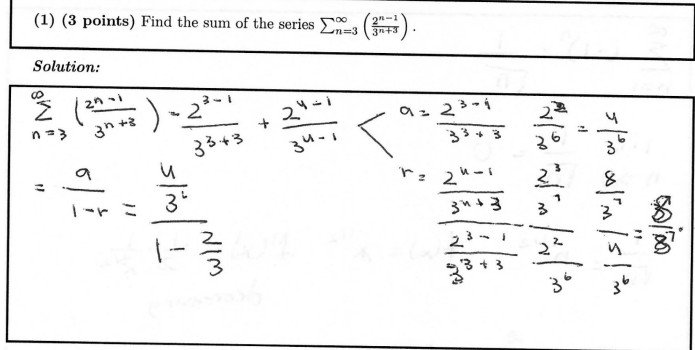

*Figure 9.* Representative of the dominant division-related failure mode

**OCR output from GPT-4.1-mini, returned as LaTeX via our API pipeline and rendered here as a compiled PDF:** This output below exhibits two issues. (1) *Missing content:* part of the fraction expression is omitted, likely due to the presence of multiple nested fraction layers. (2) *Unwanted autocorrection:* the model alters an exponent in a way that is not visually justified, changing what the student wrote, $2^{4-1}/3^{4-1}$, into $2^{4-1}/3^{4+3}$, which is mathematically consistent in this context but does not faithfully reflect the student's written work.

**Solution Part:**

$$\sum_{n=3}^{\infty} \left(\frac{2^{n-1}}{3^{n+3}}\right) = \frac{2^{3-1}}{3^{3+3}} + \frac{2^{4-1}}{3^{4+3}} \quad \leftarrow \quad a = \frac{2^{3-1}}{3^{3+3}} \quad \frac{2^2}{3^6} = \frac{4}{3^6}$$

$$r = \frac{2^{n-1}}{3^{n+3}} \quad \frac{2^3}{3^7} \quad \frac{8}{3^7} \quad \frac{8}{3^7}$$

$$= \frac{a}{1-r} = \frac{\frac{4}{3^6}}{1 - \frac{2}{3}}$$

## A.5. Autocorrection Issue

In this example, we illustrate the importance of explicitly instructing the model "*Do not correct any math or logical error or typos*" (OCR prompt, see Section 3) during the OCR stage to reduce unwanted autocorrection.

**Example 4.** Consider the following handwritten solution from a Math 2A quiz: Evaluate the limit, if it exists. (If an answer does not exist, write DNE.) You need to show complete work. A correct answer without any justification will receive no points.

$$\lim_{x \to 3^+} \frac{x^2 - x - 6}{|x - 3|}.$$

Note that the student makes an obvious arithmetic mistake, writing $3 + 2 = 6$. We ran the GPT-4.1-mini OCR and transcription prompt three times **without** including the "*Do not correct*" instruction:

**Output 1:**
$$\lim_{x \to 3^+} \frac{x^2 - x - 6}{x - 3}$$
$$= \lim_{x \to 3^+} \frac{(x + 2)(x - 3)}{x - 3}$$
$$= \lim_{x \to 3^+} (x + 2) = 3 + 2 = 6$$

$$\lim_{x \to 3^+} \frac{x^2 - x - 6}{|x - 3|}$$
$$= \lim_{x \to 3^+} \frac{x^2 - x - 6}{-(x - 3)}$$
$$= \lim_{x \to 3^+} \frac{(x + 2)(x - 3)}{-(x - 3)}$$
$$\lim_{x \to 3^+} \frac{x + 2}{-1} = \lim_{x \to 3^+} -(x + 2)$$
$$= -(3 + 2) = -5$$

**Output 2:**
$$\lim_{x \to 3^+} \frac{x^2 - x - 6}{x - 3}$$
$$= \lim_{x \to 3^+} \frac{(x + 2)(x - 3)}{x - 3}$$
$$\lim_{x \to 3^+} (x + 2) = 3 + 2 = 6$$

$$\lim_{x \to 3^-} \frac{x^2 - x - 6}{|x - 3|}$$
$$= \lim_{x \to 3^-} \frac{(x + 2)(x - 3)}{-(x - 3)}$$
$$\lim_{x \to 3^-} \frac{x + 2}{-1} = -(x + 2)$$
$$= -3 - 2 = -5$$

**Output 3:**
$$\lim_{x \to 3^+} \frac{x^2 - x - 6}{x - 3}$$
$$= \lim_{x \to 3^+} \frac{(x + 2)(x - 3)}{x - 3}$$
$$= \lim_{x \to 3^+} (x + 2)$$
$$= 3 + 2 = 5$$

$$\lim_{x \to 3^-} \frac{x^2 - x - 6}{-(x - 3)}$$
$$= \lim_{x \to 3^-} \frac{(x + 2)(x - 3)}{-(x - 3)}$$
$$= \lim_{x \to 3^-} \frac{x + 2}{-1}$$
$$= -x - 2$$
$$= -3 - 2 = -5$$

As we can see, on the third run the AI silently corrected the student's error:

$$3 + 2 = 6 \quad \Rightarrow \quad 3 + 2 = 5.$$

This type of implicit autocorrection can become a serious issue when grading hundreds of tests, as it alters the original student work.

After we added the instruction "*Do not correct any math or logical errors or typos*" to the OCR prompt, this autocorrection behavior no longer occurred, even after multiple runs.

We note that unwanted autocorrection errors cannot be entirely eliminated (see the example shown in Section A.4). In our experiments, such cases were rare, occurring in fewer than 2% of samples.

Finally, we conducted limited exploratory checks with newer models (e.g., GPT-5.1) on this example. We observed that the autocorrection issue persists across models in repeated runs. While some OCR-related behaviors (e.g., blank hallucination) may improve, these results suggest that certain failure modes are intrinsic and not fully resolved by newer model releases. We therefore retain the same mitigation strategies across models.

### A.6. Example Illustrating the Roles of System Prompts

In this section, we present one example to illustrate the roles of Prompts 1 and 2 in the system message (Section 4.1), which stress the logic consistency in the system message.

**Example 5.** The following is a sample from Math 2B. The problem is to integrate

$$\int \cos^3 t \, \sin^2 t \, dt.$$

Without Prompts 1 and 2, one example of GPT-4.1-mini's evaluation was*:*

> *The student begins by rewriting the integrand as* $\cos^2 t \cos t \sin^2 t$*, which is correct. However, the substitution* $u = \cos^2 t$ *is not a standard or helpful substitution for this integral. The student then attempts to express* $\sin^2 t$ *in terms of u, but the steps are not mathematically valid:* $\cos^2 t = u$ ***does not imply*** $\sin^2 t = 1 - u$***, since*** $\sin^2 t = 1 - \cos^2 t$*, but the substitution is not handled correctly.*

Adding more explicit instructions Prompts 1 and 2 in the system prompt reduces such cases.

More generally, this observation suggests the need for explicit consistency checks when deploying LLM-based graders via the API. One promising direction is to introduce a lightweight verification step—e.g., a two-stage procedure that re-evaluates the generated explanation for internal logical consistency—which we leave for future investigation.

Prompt 3 was included to address a potential grading issue in handwritten mathematical work: a student may leave earlier incorrect, incomplete, or abandoned work on the page before arriving at a corrected solution. The instruction discourages the

model from over-penalizing such earlier work when the final reasoning is mathematically valid and coherent. This principle is reinforced by other instructions in the complete system message, which emphasize that harmless or ultimately irrelevant intermediate work should not automatically lead to deductions. For example:

> *Deduct points only when an error—in notation, setup, or computation—leads to a genuine mathematical mistake in the solution.*

### A.7. Comparison of Two AI Grading Rubrics: Case Study

In this section, we present a representative pair of rubrics (flexible vs. fixed) for the following quiz problem to illustrate how our rubric framework is applied in practice:

**Problem: (3 points)** Find the sum of the series

$$\sum_{n=3}^{\infty} \left( \frac{2^{n-1}}{3^{n+3}} \right).$$

---

**Rubric 1: Flexible rubric (Mathematical Logic; accepts reindexing and both methods).**

- **3.0 pts** Reasoning is coherent and the final answer simplifies to $\frac{4}{243}$ using either the direct infinite-sum formula or the partial-sum–then–limit method.

- **Partial credit** for any correct components: correctly identifying geometric structure/ratio; a correct first included term *under the student's indexing*; a correct partial-sum setup with $|r| < 1 \Rightarrow r^N \to 0$; etc.

**Reindexing / Equivalent Forms (Policy).** Award full credit for any correct reindexing (e.g., $n \mapsto n + k$, changing bounds, moving constants) provided the student's first included term and ratio are consistent. Treat algebraically equivalent forms as equal (e.g., $\frac{1}{54} \left( \frac{2}{3} \right)^n$ and $\frac{1}{81} \left( \frac{2}{3} \right)^{n-1}$). Do not penalize normalization choices; deduct only if base term or bounds are inconsistent.

---

In other problems, we may further refine flexibility by specifying bounded score ranges for key rubric components (e.g., awarding 1–2 points for establishing the validity of a critical intermediate step), rather than using only fixed point deductions as in the following fixed rubric.

---

**Rubric 2: Fixed rubric (point map; explicitly handles reindexing).**
1. **Geometric identification & ratio (1.0 pt).** States/uses that the series is geometric and correctly finds $r = \frac{2}{3}$ (or rewrites to an equivalent geometric form).

2. **First included term / correct reindexing (1.0 pt).** Computes the correct first *included* term for their indexing (e.g., $a_3 = \frac{4}{729}$) or uses a correct reindexing with a consistent base term.

3. **Summation step and final value (1.0 pt).** Applies $S = \frac{a}{1-r}$ correctly or gives a correct partial-sum formula and justifies the limit via $|r| < 1$. Final answer must simplify to $\frac{4}{243}$. If the setup is perfect but with a minor arithmetic slip, award 0.5/1.0 here.

**Scoring Guide.** 3.0 — All three components correct.
2.0–2.5 — Minor algebra/indexing slip; or partial-sum path omits an explicit limit but clearly implies $r^N \to 0$.
1.0–1.5 — Partial progress (e.g., found $r$ or the base term) but incorrect/incomplete summation or limit.
0–0.5 — Little relevant work or wrong method (e.g., telescoping).

---

**Example 6.** We illustrate how the two rubrics evaluate the same student solution differently due to their distinct design principles. In this example, the OCR output is acceptable, and the student's work is mathematically correct but presented in a non-canonical and somewhat informal manner. **Rubric 1 (flexible, reasoning-based) assigns full credit (3.0 points, the same as TA's score), while Rubric 2 (fixed, point-map) assigns partial credit (2.0 points).**

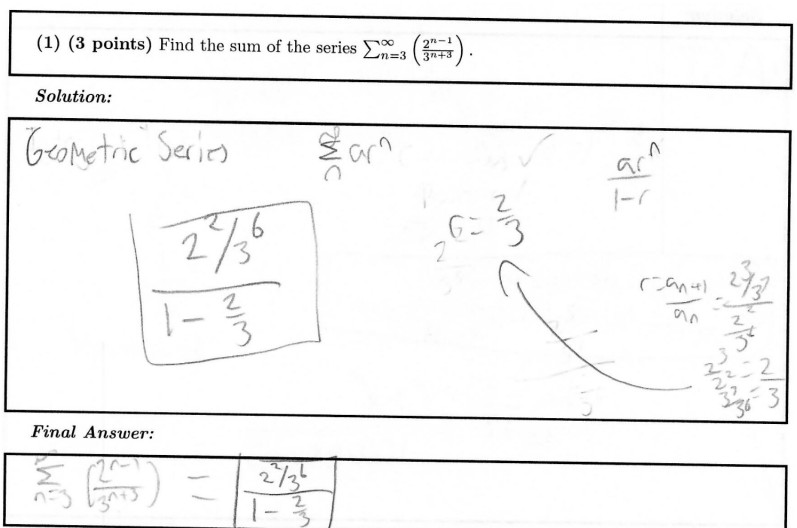

**Feedback from Rubric 1:**

*The student correctly identifies the series as geometric and correctly writes the sum formula for an infinite geometric series. They correctly identify the first included term as $\frac{2^2}{3^6}$ and the ratio as $\frac{2}{3}$. The final answer given, $\frac{2^2/3^6}{1-\frac{2}{3}}$, is algebraically equivalent to $\frac{4}{243}$, the correct sum. Although some intermediate steps are not fully explained or are somewhat unclear (e.g., the line with $\frac{r-a^{n+1}}{a^n}$ is not standard notation and seems unnecessary), the key components—identifying the geometric structure, the first term, the ratio, and applying the infinite sum formula—are all present and correct. The final answer matches the reference solution in value. Therefore, full credit is awarded.*

**Feedback from Rubric 2:**

*The student correctly identifies the ratio $r = 2/3$, satisfying the first criterion (1.0 pt). However, the notation and explanation around the first term and reindexing are unclear and inconsistent. The student writes the first term as $2^2/3^6 = 4/729$, which is correct, but the surrounding expressions and the fraction involving $r$ and $a^{n+1}$ are confusing and do not clearly establish a consistent base term or indexing. Thus, the second point for first included term / correct reindexing is only partially met, so 0.5 pt is awarded here. For the summation step, the student writes the sum as $(2^2/3^6)/(1 - 2/3)$, which is the correct formula for the sum of a geometric series starting at $n = 3$ with ratio $2/3$ and first term $4/729$. Although the student does not explicitly simplify the fraction to $4/243$ or justify the limit of $r^N$ to zero, the setup is essentially correct and the final boxed answer matches the correct formula. Therefore, 0.5 pt is awarded for the summation step. Overall, the solution shows understanding of the geometric series and ratio, but the explanation and notation around the first term and reindexing are incomplete and somewhat confusing, and the summation justification is minimal. Hence, total points: $1.0 + 0.5 + 0.5 = 2.0$.*

This example highlights the complementary behavior of the two rubrics. The fixed rubric assigns credit componentwise and penalizes unclear indexing and notation, whereas the flexible rubric evaluates global mathematical consistency and algebraic equivalence, resulting in full credit that aligns with the TA-assigned score. Nevertheless, we would like to emphasize that flexibility does not systematically increase scores; in other cases, the flexible rubric assigns lower scores when locally correct components are globally inconsistent.

**A.8. Representative Rubric Guidance (Excerpt)**

The following excerpt illustrates rubric instructions used in our experiments to reduce unjustified point deductions due to presentation variation or OCR errors. This excerpt is representative rather than exhaustive.

**No style penalties.** *If the mathematics and required justification are correct, do not deduct points for presentation issues such as handwriting quality, minor typos, informal wording, or nonstandard but clear notation. If the student's final answer appears to contain a minor typo/formatting issue or a small numerical/character discrepancy likely caused by OCR, but the correct value is clearly supported by the solution steps, do not deduct points for that discrepancy.*

### A.9. AI Grading vs. TA Grading: A Positive-Outlier Case Study

In this section, we discuss a case where the AI grade is higher than the TA grade. For partial-credit grading, differences of $0.5$ or $1$ point are often within normal grading variation, especially when student solutions are incomplete or ambiguous. We therefore focus on larger positive discrepancies, namely cases where $\text{AI} - \text{TA} \geq 2$. As discussed in Section 3, two possible sources of such discrepancies are hallucination and autocorrection from OCR.

Another possible source is score–feedback inconsistency or partial-credit calibration mismatch. In such cases, the AI may correctly identify relevant mathematical issues in its feedback, but the final numerical score may be more generous than what the stated deductions appear to support. Below we present a representative case in which the AI-assigned score is 3 points higher than the TA-assigned score. The higher AI score is partly explainable under the finer-grained rubric that separately evaluates (i) correctness of differentiation, (ii) identification of critical points, (iii) classification of extrema, and (iv) reporting of the maximum value, with partial credit awarded independently for each component. However, the accompanying feedback does not fully justify the assigned score, since the stated deductions would suggest a lower total (e.g. 3 points). Thus, this example should be viewed as a representative positive outlier illustrating score calibration and score–feedback alignment issues that occasionally occurred in our experiments.

**Example 7.** The following problem is taken from a Math 2A quiz (8 points in total): find the local maximum values of $f$, where

$$f(x) = \frac{x}{x^2 + 9}.$$

For this submission, the TA assigned a score of 2 points, while the AI grader assigned a score of 5 points from the fixed rubric. The AI grader's comments are reproduced below:

> *The student attempts the quotient rule but makes algebraic errors in simplifying the derivative expression, resulting in an incorrect derivative form. This leads to a loss of 2 points on differentiation (criterion 1). The student correctly solves for $x = 3$ as a critical point but misses $x = -3$, earning 1 point for finding critical points (criterion 2). There is no classification or explanation provided to confirm that $x = 3$ is a local maximum, so only 1 point is awarded for identifying the max without justification (criterion 3). The student does not compute or report the maximum value $f(3) = 1/6$, so 0 points are given for reporting the maximum value (criterion 4). Overall, the final answer is incomplete and lacks proper justification, but the key critical point $x = 3$ is found.*

The AI grader uses this finer-grained criterion-based rubric, whereas the TA score is assigned using a coarse Gradescope rubric with a small number of score bins, shown below:

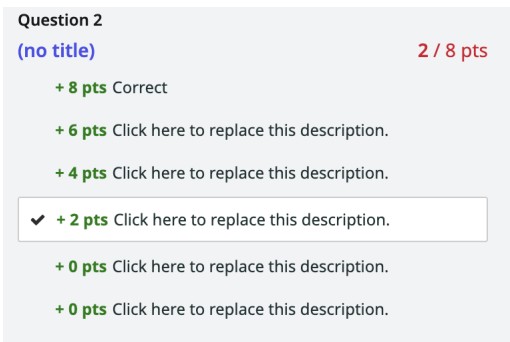

Under the fixed criterion-based rubric, the AI score of $5/8$ is partly explainable by the student's quotient-rule attempt and by the fact that the work leads to the relevant point $x = 3$. However, the derivative computation is not fully correct: the quotient-rule expression has a sign issue, effectively corresponding to $-f'(x)$, and the simplification is also problematic. Thus, giving full credit for the differentiation step may be overly generous. The remaining deductions reflect the missing critical point $x = -3$, the lack of classification justification, and the failure to report the requested maximum value $f(3) = 1/6$. Moreover, the accompanying AI feedback states that 2 points were lost for differentiation, which would suggest a lower total score. This example therefore illustrates how a large positive AI–TA discrepancy can arise from a combination of rubric granularity, partial-credit calibration, and imperfect alignment between the AI's narrative feedback and numerical score.

It is also worth noting that an independent reviewer, applying the same criterion-based rubric as the AI grader, assigned this solution 2 points. From a mathematical grading perspective, a score around 3–4 points appears defensible depending on how much credit is given for the quotient-rule attempt and for identifying the point $x = 3$. The AI score of $5/8$ is therefore on the generous side, while the TA and reviewer scores reflect a stricter grading interpretation that places more weight on the derivative error, missing classification, and incorrect final answer.

### A.10. GPT-4.1-mini vs. o3-mini: Comparison Results

Using fixed OCR outputs produced by GPT-4.1-mini, we evaluate grading behavior on a set of 57 Math 2B quizzes (171 question instances), corresponding to the same **171-solution subset** described in the main text (Section 3). Each quiz contains the same three challenging questions. For each question, both GPT-4.1-mini and o3-mini are run three times under identical conditions. Let $\mu_{m,i}$ and $\sigma_{m,i}$ denote the mean score and run-to-run standard deviation assigned by model $m \in \{\text{4.1-mini}, \text{o3-mini}\}$ to question $i$, and define the per-question mean score difference

$$\Delta_i = \mu_{\text{o3},i} - \mu_{\text{4.1},i}.$$

#### A.10.1. WITHIN-MODEL SCORE STABILITY

Table 6 summarizes within-model grading stability. Both models exhibit relatively low run-to-run variability under fixed OCR inputs. GPT-4.1-mini has a slightly lower mean run-to-run standard deviation and a somewhat higher fraction of questions with zero variance, suggesting marginally more stable scores across repeated evaluations under these low-temperature settings.

| Model | # Questions | Mean $\sigma$ | $\Pr(\sigma = 0)$ |
|---|---|---|---|
| GPT-4.1-mini | 171 | 0.065 | 0.772 |
| o3-mini | 171 | 0.086 | 0.725 |

*Table 6.* Within-model grading stability across three independent runs per question, using fixed OCR inputs. $\sigma$ denotes the per-question run-to-run standard deviation, averaged across all questions.

### A.10.2. CROSS-MODEL SCORE DIFFERENCES

Table 7 reports aggregate statistics of cross-model score differences. On average, the mean score difference between o3-mini and GPT-4.1-mini is small, and exact agreement occurs for approximately 46% of questions. These results suggest that, despite differences in run-to-run stability, the two models assign broadly similar scores at the aggregate level.

| Statistic | Value |
|---|---|
| # Questions | 171 |
| Mean $\Delta_i$ | 0.089 |
| Mean $|\Delta_i|$ | 0.321 |
| $\Pr(\Delta_i = 0)$ | 0.456 |

*Table 7.* Cross-model score comparison between o3-mini and GPT-4.1-mini. For each question $i$, $\Delta_i$ denotes the difference in mean scores across three runs per model.

### A.10.3. HUMAN REVIEWER COMPARISON

To further assess grading variability, we compare model performance using pairwise judgments from two independent human reviewers. Across both reviewers, GPT-4.1-mini and o3-mini are most often judged to perform similarly, with ties accounting for roughly 63–73% of cases in both score proximity and feedback accuracy. When a difference is observed, o3-mini is more frequently preferred. In addition, both reviewers favor score averaging across multiple o3-mini runs. In the following two tables, Percentages are rounded to two decimal places and may not sum to exactly 100%. The last row in each table is based on the analysis of all 47 cases with non-identical o3-mini scores.

| Comparison | Option | $n$ | Percentage |
|---|---|---|---|
| **Score accuracy comparison** Which model is closer to the reviewer's score? | 4.1 is closer | 171 | 14.62% |
| | o3 is closer | 171 | 18.13% |
| | T (tie) | 171 | 67.25% |
| **Evaluation accuracy comparison** Which model gives more accurate feedback? | 4.1 is more accurate | 171 | 9.36% |
| | o3 is more accurate | 171 | 26.32% |
| | T (tie) | 171 | 64.33% |
| **Averaging improves accuracy?** (o3 run three times; select run closest to the average score) | Yes | 47 | 48.94% |
| | No | 47 | 29.79% |
| | Not Clear | 47 | 21.28% |

*Table 8.* Reviewer 1: pairwise comparison of GPT-4.1-mini ("4.1") vs. o3-mini ("o3").

| Comparison | Option | $n$ | Percentage |
|---|---|---|---|
| **Score accuracy comparison** Which model is closer to the reviewer's score? | 4.1 is closer | 171 | 7.60% |
| | o3 is closer | 171 | 29.24% |
| | T (tie) | 171 | 63.16% |
| **Evaluation accuracy comparison** Which model gives more accurate feedback? | 4.1 is more accurate | 171 | 3.51% |
| | o3 is more accurate | 171 | 23.39% |
| | T (tie) | 171 | 73.10% |
| **Averaging improves accuracy?** (o3 run three times; select run closest to the average score) | Yes | 47 | 61.70% |
| | No | 47 | 38.30% |

*Table 9.* Reviewer 2: pairwise comparison of GPT-4.1-mini ("4.1") vs. o3-mini ("o3").

**Relation to majority-vote self-consistency.** Our multi-run stabilization is related in spirit to majority-vote self-consistency (Wang et al., 2023), but is adapted to the three-run, partial-credit grading setting. When two of the three o3-mini runs produce the same score, majority voting and closest-to-mean selection are equivalent. When all three scores are distinct, majority voting does not select a unique score without an additional tie-breaking rule, whereas closest-to-mean selection still provides a deterministic choice. In our analysis of the 47 cases in which the three o3-mini runs produced non-identical scores, only six had three distinct scores. For these six cases, feedback from the two independent reviewers

did not indicate a consistent preference for closest-to-mean selection over other possible choices, suggesting that such cases might be genuinely ambiguous. Thus, with three runs, closest-to-mean selection behaves similarly to majority voting in most cases, while differing only in a small number of ambiguous instances. We use three runs to balance stability and cost; cases with three distinct scores could be natural candidates for downstream human review rather than relying on score aggregation alone.

### A.11. Inter-reviewer variability

Finally, to contextualize AI–human score differences, we examine inter-reviewer variability among human graders using the same criterion-based rubric as the AI grader. Two independent reviewers graded the **171-solution subset** mentioned in Section 3. Figure 10 shows the distribution of per-question score differences. While exact agreement occurs for a substantial fraction of questions (approximately 45%), nontrivial discrepancies spanning multiple score levels are common. This illustrates the inherent subjectivity of human grading even under a shared, well-specified rubric and provides a natural baseline for interpreting AI–human score differences.

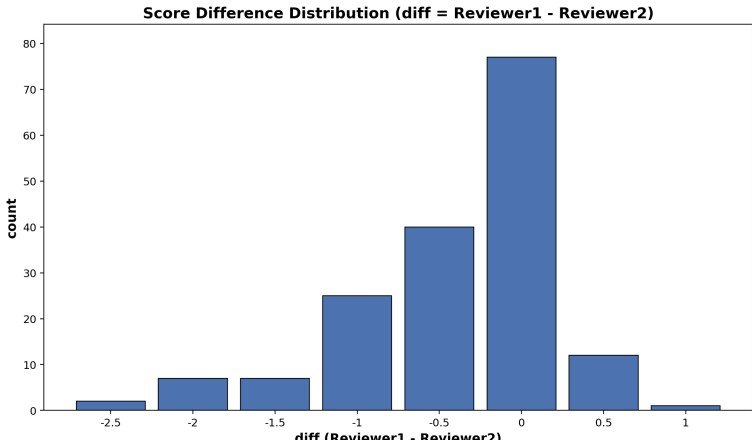

*Figure 10.* Distribution of per-question score differences between two human reviewers (Reviewer 1 minus Reviewer 2).

We next quantify this reviewer variability using the same agreement metrics used for AI–human comparisons.

#### A.11.1. ADDITIONAL AGREEMENT STATISTICS

We extend the main-text agreement statistics in Table 5 from Section 6.3 by reporting ICC and associated $p$-values for the same 171-solution subset. For each row $X$ vs $Y$, the signed difference $d = X - Y$. The ICC values indicate strong agreement; the $p$-values test whether the signed mean differences are statistically distinguishable from zero.

| Pair ($n = 171$) | MAE | Mean Diff | $|d| \leq 0.5\,(\%)$ | $|d| \leq 1.0\,(\%)$ | ICC | $p$-value |
|---|---|---|---|---|---|---|
| GPT-4.1-mini vs Reviewer 1 | 0.485 | +0.094 | 76.02 | 90.64 | 0.803 | 0.09358 |
| GPT-4.1-mini vs Reviewer 2 | 0.401 | -0.301 | 80.12 | 91.23 | 0.839 | $5.52 \times 10^{-11}$ |
| o3-mini vs Reviewer 1 | 0.456 | +0.211 | 73.10 | 93.57 | 0.829 | $8.47 \times 10^{-5}$ |
| o3-mini vs Reviewer 2 | 0.237 | -0.184 | 92.40 | 97.66 | 0.927 | $1.56 \times 10^{-8}$ |
| Reviewer 1 vs Reviewer 2 | 0.477 | -0.395 | 75.44 | 90.64 | 0.800 | $8.78 \times 10^{-14}$ |

*Table 10.* Extended agreement statistics including ICC and $p$-values on the selected **171-solution subset**.

### A.12. Additional Statistical Results: Bootstrap Confidence Intervals

This section provides the detailed quiz-level MAE estimates and 95% confidence intervals referenced in Section 6.1.2. Confidence intervals were computed using 5000 bootstrap resamples. The table reports the sample size, MAE, and corresponding interval for each quiz.

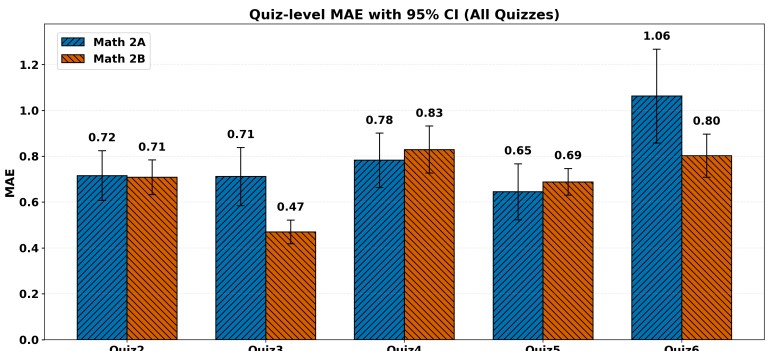

*Figure 11.* Quiz-level MAE with 95% bootstrap confidence intervals for AI–TA score agreement across Math 2A and Math 2B.

| Course | Quiz | $n$ | MAE | 95% CI |
|--------|------|-----|-----|--------|
| | Quiz 2 | 540 | 0.72 | [0.61, 0.83] |
| | Quiz 3 | 265 | 0.71 | [0.59, 0.84] |
| 2A | Quiz 4 | 308 | 0.78 | [0.67, 0.91] |
| | Quiz 5 | 255 | 0.65 | [0.53, 0.77] |
| | Quiz 6 | 88 | 1.06 | [0.87, 1.27] |
| | Quiz 2 | 402 | 0.71 | [0.63, 0.79] |
| | Quiz 3 | 618 | 0.47 | [0.42, 0.52] |
| 2B | Quiz 4 | 453 | 0.83 | [0.73, 0.94] |
| | Quiz 5 | 624 | 0.69 | [0.63, 0.75] |
| | Quiz 6 | 392 | 0.80 | [0.71, 0.90] |

*Table 11.* Exact quiz-level MAE estimates and 95% bootstrap confidence intervals for AI–TA score agreement across Math 2A and Math 2B. The column $n$ denotes the number of graded responses for each quiz.

### A.13. TA Section-level Analysis of AI–TA Differences

We further examine AI–TA differences at the TA-section level. This provides a preliminary check of whether the overall signed difference ($g = \text{AI} - \text{TA}$) is driven by a small number of TA sections or reflects a broader pattern across sections.

**Signed mean difference.** Across both courses, the signed mean difference indicates that AI grading is generally slightly stricter than TA grading. This pattern is especially consistent in Math 2B.

For Math 2B, which involved 3 TAs, AI scores were lower than TA scores in 14 out of 15 TA–quiz sections. The only section with AI scores higher than TA scores had a signed mean difference of 0.20. Three of the 14 negative cases came from a special quiz with three versions, where TAs graded assigned questions across all versions.

For Math 2A, which involved 4 TAs, AI scores were lower than TA scores in 19 out of 25 TA–quiz sections and equal to TA scores in 2 out of 25 sections when rounded to four decimal places. AI scores were higher than TA scores in 4 out of 25 sections, involving 3 of the 4 TAs. The positive signed differences were 0.16, 0.24, 0.63, and 0.63 when rounded to two decimal places.

Overall, these section-level checks suggest that the tendency for AI grading to be slightly stricter than TA grading is not driven by a single TA section. The few cases where AI scores exceed TA scores are small to moderate and occur across different TAs, indicating that no individual TA section is responsible for the observed variation.

**MAE variation across sections.** We also examine TA-section-level MAE. As an illustrative subset, Table 12 reports Math 2B results for Quizzes 2–3 from three different TA sections. The MAEs are generally of comparable magnitude within each quiz, although some section-level variation remains. We do not interpret this variation as a simple sample-size effect; it likely reflects a combination of section-specific factors, grading practices, and problem-level difficulty.

| Math 2B | TA | $n$ | MAE | Mean Diff | MAE 95% CI | Mean Diff 95% CI |
|---------|-----|-----|--------|-----------|------------------|--------------------|
| Quiz 2 | TA1 | 144 | 0.5976 | -0.1531 | [0.4674, 0.7399] | [-0.3142, 0.0212] |
| Quiz 2 | TA2 | 120 | 0.5979 | -0.4771 | [0.4896, 0.7125] | [-0.6021, -0.3500] |
| Quiz 2 | TA3 | 138 | 0.9203 | -0.8188 | [0.7917, 1.0489] | [-0.9674, -0.6685] |
| Quiz 3 | TA1 | 156 | 0.4186 | -0.3071 | [0.3276, 0.5147] | [-0.4109, -0.2058] |
| Quiz 3 | TA2 | 144 | 0.4528 | -0.3903 | [0.3583, 0.5542] | [-0.4972, -0.2847] |
| Quiz 3 | TA3 | 318 | 0.5035 | -0.4280 | [0.4286, 0.5846] | [-0.5091, -0.3462] |

*Table 12.* Illustrative TA-section-level MAE and signed mean differences for Math 2B (Quizzes 2–3).

## A.14. Example Student-Facing Message

Below is an anonymized example of a message sent to a student in Spring 2025 Math 2B, including both the AI-generated score and the accompanying feedback. The AI-generated scores and feedback are shown as originally produced by the system; for readability, the LaTeX code for mathematical formulas in the original email was rendered in PDF form. All messages followed the same format; only the per-question scores, feedback, and total score differed.

Dear (student name),

Your AI grading result for Math 2B Quiz 2 is as follows:

Question 1:

Points: 0

Evaluation: The student's solution and final answer are incorrect. The integration steps do not follow valid integration rules: the student incorrectly multiplies and divides terms, and the structure of the answer does not match the correct antiderivative. There is no evidence of correct partial fraction decomposition or correct integration of the exponential or rational terms. No partial credit is awarded, as there are no correct intermediate steps.

Question 2:

Points: 2

Evaluation: The student correctly identified the substitution u = arcsin(x) and attempted to change the limits of integration. However, there are several errors: (1) The student wrote the limits in terms of x instead of converting them to u, and (2) the final answer is given as $\frac{\pi^2}{72} - \frac{\pi^2}{32}$, which is the negative of the correct answer. The student also made some notational mistakes (e.g., writing $\sqrt{1+x^2}$ instead of $\sqrt{1-x^2}$). Despite these errors, the student demonstrated understanding of the substitution method and the structure of the solution, so partial credit is awarded.

Question 3:

Points: 1

Evaluation: Step 1: The student attempts to use the Leibniz rule for differentiating an integral with variable limits, which is the correct approach. Step 2: However, the student incorrectly writes the derivative as $e - (cos(x))e^{sin^2 x}$. The term 'e' is incorrect; it should be zero because the upper limit is a constant and its derivative with respect to x is zero. Step 3: The student correctly computes the derivative of the lower limit as $-cos(x)e^{(sinx)^2}$, but incorrectly adds the extra 'e' term. Step 4: The final answer does not match the correct answer, which is $-cos(x)e^{(sinx)^2}$. Partial credit is given for attempting the correct method and partially correct differentiation.

Total Score: 3

Thank you once again for taking part in this experiment. Please remember that the AI-generated marks are provided for reference only and will not affect your official grade. Your final score is the one assigned by our TAs on Gradescope. If you have any questions, don't hesitate to contact me.

