# OpenReview forum: "Evaluating AI Grading on Real-World Handwritten College Mathematics: A Large-Scale Study Toward a Benchmark"
_ICML.cc/2026/Conference — ICML 2026 regular_

### Official Review · Reviewer_4h6W · 2026-03-11

**Soundness:** 2
**Presentation:** 3
**Significance:** 4
**Originality:** 2
**Overall Recommendation:** 4
**Confidence:** 3

**Summary:**

This paper presents a large-scale empirical study of AI-assisted grading for handwritten single-variable calculus at a major U.S. public research university. The system combines OCR (GPT-4.1-mini) with LLM-based rubric-guided prompting to produce scores and formative feedback for thousands of free-response quiz submissions from nearly 800 students across three academic terms. A dual-rubric design (flexible + fixed) with a max-rule aggregation strategy is proposed. Since no ground-truth labels exist, evaluation is conducted via three complementary lenses: alignment with TA scores (3,945 records), student surveys (~300 respondents), and independent human review by 20+ reviewers (3,851 records). The authors report strong AI-TA alignment (MAE 0.5--1.06, within-1 rate 68--86%), ~90% correct-or-acceptable feedback per independent review, and outline plans for a future benchmark release.

**Compliance With Llm Reviewing Policy:**

Affirmed.

**Final Justification:**

The rebuttal and follow-up reply have improved my view of the paper. W1 (circular bias) is resolved — the proctored setting is a valid response. W3 (stats) is well addressed with bootstrap CIs and t-tests. W5 (max-rule) — the breakdown showing AI > TA in only ~12% of cases is convincing enough. W4 (reproducibility) — the authors confirmed that released artifacts will allow full reproduction without re-running proprietary models, which is reasonable.

On W6 (fairness), the newly added TA-section-level analysis is a meaningful step. The variation across sections (e.g., MAE ranging from ~0.42 to ~0.92 within the same quiz) does suggest section-specific factors worth investigating, but the fact that the authors ran this analysis in response to review feedback is appreciated.

W2 (benchmark) remains a limitation — the benchmark is still not available. But the title does say "toward," and the empirical study itself carries value independently.

Overall, this is a solid applied study at a rare real-world scale. The main weaknesses I flagged have been either addressed or acknowledged with concrete plans. I'm raising my score accordingly.

**Key Questions For Authors:**

1. **Circular bias / AI-generated student work:** Did you investigate or attempt to detect whether any student submissions were generated or heavily influenced by LLMs? If students copy ChatGPT solutions, the AI grader may systematically favor those answers due to shared stylistic and structural patterns. How would you address this risk, and does it affect the interpretation of your AI-TA alignment results (e.g., could high alignment partly reflect that both AI and TAs give full marks to ChatGPT-quality canonical solutions)?

2. **Max-rule inflation:** By construction, max(flexible, fixed) ≥ each individual rubric score. Have you measured how often the max-rule produces scores *above* the TA score? Is the reported alignment partly a consequence of the max-rule inflating scores to match TAs who may themselves be lenient?

3. **Survey self-selection:** The ~40% response rate raises concerns about selection bias. Did you compare the AI-TA score gap distribution for survey respondents vs. non-respondents? Students who received unfavorable AI scores may be more (or less) likely to respond.

4. **Demographic fairness:** Have you analyzed whether OCR error rates or score gaps vary across student subgroups (e.g., by handwriting quality, native language, or TA section)? The system's acknowledged bias against messy handwriting could disproportionately affect specific populations.

5. **Rubric construction cost:** You mention rubrics are "iteratively refined" with "light human editing" for new problems. Can you quantify the human effort (person-hours) required per quiz? This is crucial for assessing whether the system genuinely reduces TA workload or merely shifts it to rubric engineering.

6. **Why not evaluate on proof-based or multi-step problems** beyond computational calculus? The generalizability claim would be substantially stronger with even a small pilot on a different mathematical domain.

**Limitations:**

yes

**Strengths And Weaknesses:**

## Strengths

1. **Rare real-world scale.** The study deploys an end-to-end AI grading system in a live, for-credit university course over three semesters. This scale and ecological validity are uncommon in the AI grading literature, which is often limited to small, retrospective evaluations.

2. **Thoughtful multi-perspective evaluation.** In the absence of ground truth, the three-pronged evaluation (TA alignment, student feedback, independent review) is a pragmatic and well-motivated design. Each perspective addresses a different facet of reliability.

3. **Detailed and honest failure analysis.** The paper provides concrete OCR failure modes (nested fractions, crossed-out text, diagrams), rubric edge cases (non-canonical solutions, style-based penalties), and system-level failure patterns. The appendix contains numerous illustrative examples that make the analysis tangible.

4. **Practical engineering insights.** The dual-rubric + max-rule design, the "do not correct errors" OCR prompt, the system-message refinements to prevent logical inconsistency and false-start penalization, and the multi-run stabilization for o3-mini are all useful, transferable design patterns for practitioners.

5. **Transparent writing.** The authors are candid about limitations, including the lack of a systematic intervention/deferral policy, single-institution scope, and incomplete rubric aggregation comparisons.

---

## Weaknesses

### W1. Missing analysis of LLM-generated student work and circular bias

This is perhaps the most significant gap. The paper covers 2025 data, during which students had widespread access to ChatGPT and similar tools. Yet there is **no discussion** of the possibility that students may have used LLMs to generate or heavily guide their solutions, nor of the resulting fairness implications. This creates a plausible circular bias:

- **OCR stage:** LLM-copied solutions tend to be more structured and legible, leading to fewer OCR errors and thus fewer unjustified point deductions. Meanwhile, students writing original work with messier handwriting bear the brunt of the system's acknowledged −0.40 mean score bias.
- **Rubric alignment stage:** ChatGPT-generated solutions follow canonical solution paths, naturally aligning with both the fixed rubric's step-by-step checklist and the flexible rubric's reasoning-level evaluation. The max-rule would then consistently yield high scores for such submissions.
- **LLM self-preference:** The paper itself notes (Section 6.3) that a failure mode is "undue leniency toward rubric-shaped explanations that follow familiar steps but contain incorrect reasoning." LLM-generated answers are precisely the type most likely to exhibit this rubric-shaped structure, even when flawed.

This circular bias could systematically **reward students who copy from AI** while **penalizing students who produce original handwritten work**---directly undermining the fairness claims central to the paper.

### W2. Benchmark is aspirational, not delivered

The title includes "Toward a Benchmark," and the fourth main contribution describes benchmark plans in detail (Track A clean, Track B noisy). However, no benchmark data, code, or evaluation protocol is actually released or made available for review. This substantially weakens one of the four stated contributions.

### W3. Statistical rigor of evaluation

- No confidence intervals, bootstrap estimates, or significance tests accompany any of the reported metrics (MAE, within-1 rates, feedback accuracy percentages).
- The student survey has a ~40% response rate with voluntary participation, introducing significant self-selection bias that is not discussed.
- The inter-reviewer agreement analysis (Appendix A.10.4) is limited to a single histogram; no standard inter-rater reliability statistics (e.g., Cohen's kappa, ICC) are reported.

### W4. Reproducibility concerns

The entire system depends on proprietary, closed-source models (GPT-4.1-mini, o3-mini) accessed via commercial APIs. These models may be updated, deprecated, or behave differently over time. No open-source alternatives are evaluated, and no model artifacts are released.

### W5. Limited scope of the max-rule evaluation

The max-rule is the only aggregation strategy compared against the two individual rubrics (Figure 2). The authors acknowledge in the Limitations section that averaging and other monotone transforms were not tested. Additionally, there is no analysis of whether the max-rule systematically inflates scores (it is a one-sided operator by construction) and whether this inflation differentially benefits certain solution styles.

### W6. Systematic bias against certain student populations

The −0.40 mean score bias and the concentration of negative gaps in OCR errors imply that students with poor handwriting, non-standard notation, or limited English proficiency (affecting written explanations) are systematically disadvantaged. The paper acknowledges OCR-related deductions but does not analyze whether these errors are correlated with demographic or educational factors, which is critical for any fairness claim.

---

> ### Author Rebuttal · Authors · 2026-03-31
>
> We thank the reviewer for the detailed and constructive feedback. We address the main concerns below.
>
> **1. Circular bias / LLM-generated student work**
>
> Our system is designed for proctored, in-person assessments (quizzes, midterms, finals), where students produce handwritten solutions **without** LLM assistance (**only** pen and scratch paper are allowed).
>
> All data come from such controlled settings. Therefore, the **circular-bias scenario** described by the reviewer is **outside the intended deployment setting of this work** and does not affect the reported results directly. We agree, however, that for take-home or AI-assisted settings this would become a serious fairness concern and should be studied explicitly.
>
> We also acknowledge a **subtler fairness concern**: even without direct LLM access during the assessment, some students may learn to present solutions in ways that better match rubric- or model-preferred patterns, which could create unintended advantages and deserves further study.
>
> - This type of undue leniency may be mitigated in part by stronger reasoning models (e.g. o3-mini), which are better able to detect inconsistencies in otherwise well-structured solutions.
> - In addition, closely monitoring grading behavior and periodically refining prompts and rubrics may be necessary to reduce such effects in practice.
>
> We will include this issue in the revision.
>
> ---
>
> **2. Benchmark availability**
>
> "Toward a benchmark" refers to establishing the dataset, evaluation protocol, and construction pipeline. Most core components are already constructed and used internally; public release is pending privacy/compliance review for handwritten data and final expert verification.
>
>
> **3. Statistical rigor**
>
> Our focus is **deployment-oriented reliability** (MAE, within-1 rate, independent review). Following the reviewer’s suggestion, we add **95% CIs** (computed via bootstrap with 5,000 resamples):
>
> - **95% CI MAE:**
>
> | Course | Quiz  | n_pairs | MAE   | 95% CI_low | 95% CI_high |
> |--------|-------|--------:|------:|----------:|-----------:|
> | A      | Quiz2 |     540 | 0.716 |     0.611 |      0.829 |
> | A      | Quiz3 |     265 | 0.711 |     0.591 |      0.840 |
> | A      | Quiz4 |     308 | 0.782 |     0.666 |      0.906 |
> | A      | Quiz5 |     255 | 0.645 |     0.527 |      0.771 |
> | A      | Quiz6 |      88 | 1.063 |     0.869 |      1.273 |
> | B      | Quiz2 |     402 | 0.708 |     0.635 |      0.786 |
> | B      | Quiz3 |     618 | 0.470 |     0.419 |      0.522 |
> | B      | Quiz4 |     453 | 0.829 |     0.727 |      0.936 |
> | B      | Quiz5 |     624 | 0.688 |     0.631 |      0.747 |
> | B      | Quiz6 |     392 | 0.802 |     0.709 |      0.899 |
>
> - **95% CI Within-1:**  (e.g., A-quiz6: 68.2% [58.0, 78.4]; B-Quiz3: 85.6% [82.7, 88.3])
>
> - **Independent review:** feedback accuracy with 95% CIs (n=3851):
>   **Correct**: 79.79% [78.47, 81.02],
>   **Acceptable**: 9.55% [8.65, 10.46],
>   **Incorrect**: 10.67% [9.71, 11.69]
>
> We will include these in the revision.
>
> ---
>
> **4. Max-rule and potential inflation**
>
> The max-rule captures **diverse valid reasoning** (fixed rubric for canonical solutions; flexible rubric for non-standard correct approaches).
>
> Empirically, it does **not introduce upward bias**:
> AI scores are on average **0.40 points lower than TA scores**, with
> AI < TA: 39.70%, AI = TA: 48.44%, AI > TA: 11.86%.
>
> It also achieves **equal or lower MAE** than either rubric alone, supporting it as a practical aggregation strategy rather than a source of inflation.
>
> ---
>
> **5. Reproducibility**
>
> Reproducibility concerns **protocol rather than exact outputs**. We will release benchmark data, OCR outputs, rubrics, prompts, model outputs, and evaluation scripts.
>
> We acknowledge reliance on proprietary models as a limitation and will clarify this.
>
> ---
>
> **6. Fairness**
>
> We agree subgroup-level fairness is important. While OCR errors may affect certain students, we have **not yet conducted systematic analysis**. We will clarify this limitation and connect it to future **human-in-the-loop deferral policies**.
>
> ---
>
> **7. TA effort**
>
> Rubrics are generated via templates:
>
> - Step 1: GPT-o3-mini draft
> - Step 2: light human edit
>
> Based on our Fall 2025 validation experience, this takes **~15–20 minutes per problem**, indicating modest overhead.
>
> ---
>
> **8. Scope**
>
> We focus on **computational calculus** as a realistic large-scale deployment setting. Extending to proof-based problems is an important future work.
>
> ---
>
> **9. Survey limitations**
>
> The survey is anonymous, so individual linkage is not possible. After correcting the course attribution (see below)， the combined response rate from course A **approximately 75% response rate (301/~400)** helps mitigate non-response bias.
>
> ---
>
> **Correction (for all reviewers)**
> The survey data come from two sections of Course A (not A vs. B); this affects interpretation but not results.

---

> > ### Author Rebuttal · Reviewer_4h6W · 2026-04-03
> >
> > Thanks for the rebuttal.
> >
> > **Resolved:** W3 (statistical rigor) is addressed well with the bootstrap CIs and paired t-tests. W1 (circular bias) — the proctored setting clarification is fair, I accept this. W5 (max-rule) — the empirical numbers (AI > TA only 11.86%) are reasonable.
> >
> > **Remaining:** W2 (benchmark) — still not released; I'd suggest clearly separating delivered vs. planned contributions in the revision. W6 (fairness) — no subgroup analysis was done. Even without demographic data, proxy analyses (e.g., by OCR quality or TA section) would help.
> >
> > **Follow-up on W4:** Will the released artifacts (OCR outputs, model outputs, scripts, etc.) be sufficient to reproduce the main results without re-running the proprietary models?

---

> > > ### Author Response · Authors · 2026-04-05
> > >
> > > We thank the reviewer for the constructive questions and comments. We address the follow-up points below.
> > >
> > > **(1) Benchmark framing W2**
> > >
> > > Thank you for pointing this out. In the revision, we will clearly separate delivered components from planned releases and position the work explicitly as *toward a benchmark*.
> > >
> > > **(2) Fairness W6**
> > >
> > > We agree that fairness is an important direction. Following the reviewer’s suggestion, we conducted preliminary proxy analyses and outline below several next steps.
> > >
> > > ### **AI vs. TA across TA sections**
> > >
> > > As a **preliminary section-level** check, we examined both the **Course A** and **Course B** quizzes by **TA-sections**.
> > >
> > > **1. Signed mean difference: AI-TA.**
> > >
> > > Across the two courses, the **signed mean difference** indicates an overall tendency for **AI** grading to be slightly stricter than **TA** grading, with this pattern more consistent in **Course B** than in **Course A**.
> > >
> > > - **Course  B (3 TAs)**
> > >
> > >     - **AI<TA**  in  **14 out  of 15** TA-quiz sections. Three of these 14 cases come from a special quiz with three versions, where the TAs graded assigned questions across all versions.
> > >
> > >     - **AI>TA** in **one** TA-quiz section. The difference is positive **0.20**.
> > >
> > > - **Course  A (4 TAs)**
> > >
> > >     - **AI<TA** in  **19** out of **25** TA-quiz sections,
> > >
> > >     - **AI=TA** in **2** out of **25** TA-quiz sections (when rounded to four decimal places).
> > >
> > >     - **AI>TA** in **4** out of **25** TA-quiz sections, involving **3 out of 4 TAs**.   The positive differences are  **0.16, 0.24, 0.63, 0.63** (when rounded to two decimal places)
> > >
> > > These preliminary TA-section-level checks suggest that the overall tendency (**AI slightly stricter than TA**) is not driven by a single TA section. The few cases where **AI > TA** are small to moderate (**≤0.63**) and involve **different** TAs, indicating that no single TA-section is responsible for the observed variation.
> > >
> > > **2. MAE**
> > >
> > > The TA-section-level **MAEs** are generally of similar magnitude within each quiz, though some section-level variation remains.  For illustration, we report a subset of results from **Course B** below:
> > > | Course B | TA-Section | n_pairs | MAE | Mean Diff | MAE 95% CI | Mean Diff 95% CI |
> > > |--------|---------|-----|--------|--------------------|--------------------|--------------------|
> > > | Quiz 2 | TA1 | 144 | 0.5976 | -0.1531 | [0.4674, 0.7399] | [-0.3142, +0.0212] |
> > > | Quiz 2 | TA2 | 120 | 0.5979 | -0.4771 | [0.4896, 0.7125] | [-0.6021, -0.3500] |
> > > | Quiz 2 | TA3 | 138 | 0.9203 | -0.8188 | [0.7917, 1.0489] | [-0.9674, -0.6685] |
> > > | Quiz 3 | TA1 | 156 | 0.4186 | -0.3071 | [0.3276, 0.5147] | [-0.4109, -0.2058] |
> > > | Quiz 3 | TA2 | 144 | 0.4528 | -0.3903 | [0.3583, 0.5542] | [-0.4972, -0.2847] |
> > > | Quiz 3 | TA3 | 318 | 0.5035 | -0.4280 | [0.4286, 0.5846] | [-0.5091, -0.3462] |
> > >
> > > We do not interpret this variation as a simple sample-size effect; it more likely reflects a combination of section-specific factors that would require further analysis to disentangle.
> > >
> > > **(3) Reproducibility (follow-up on W4)**
> > >
> > > **Yes.** The released artifacts—including OCR outputs, model-generated scores, anonymized TA and independent-reviewer labels, OCR evaluation labels, feedback evaluation labels, and evaluation scripts—will be sufficient to **reproduce all** reported metrics and statistical results in the paper and rebuttal, including score agreement metrics (e.g., MAE, within-1), OCR evaluation results, feedback accuracy rates, bootstrap confidence intervals, and paired-comparison results, **without re-running proprietary models**.
> > >
> > > Public release of the original handwritten samples, which would further improve transparency, would require additional privacy/compliance clearance, especially if paired with TA labels.

---

### Official Review · Reviewer_fRQp · 2026-03-13

**Soundness:** 4
**Presentation:** 3
**Significance:** 4
**Originality:** 3
**Overall Recommendation:** 5
**Confidence:** 5

**Summary:**

Proposes a workflow for llm-augmented automated evaluation pipeline of Calculus quizzes incorporating OCR, and rubric-based evaluations. Reports on large scale trials over two semesters. Discusses methods and refinements, and suggests avenues for improvement including error detection and human-in-the-loop interventions. Further, the authors propose a benchmark for future work.

**Compliance With Llm Reviewing Policy:**

Affirmed.

**Final Justification:**

On review, I find my initial assessment strengthened and find the overall soundness improved.

**Key Questions For Authors:**

1- Did you try any other selection rules? Is there an inherent bias in always choosing the max?
2- How were exemplars selected for rubric generation?
3- This is not very clear to me: 6.1.2 “cross quizzes, the mean absolute gap (MAE) typically falls in the range 0.5 and 1.06 points, and the within-1 rate ranges from 68% to 86%, demonstrating strong overall alignment with TA grading.” Paired-t tests with significance intervals are more commonly used.

**Limitations:**

Yes

**Strengths And Weaknesses:**

The authors present a robust system based on commercially available models (gpt-4.1-mini) and report empirically grounded findings based on real-world tests. Their pipeline shows strong correlation with human assessments, based on an iterative design methodology with lessons learned. One shortcoming is some unconventional choices of statistics and the lack of significance tests.

---

> ### Author Rebuttal · Authors · 2026-03-31
>
> We thank the reviewer for the thoughtful and technically detailed feedback. We appreciate the focus on statistical rigor and evaluation methodology, which align closely with the goals of this work, and have strengthened the analysis accordingly.
>
>
> **1. Statistics-related questions.**
>
> Our goal is to evaluate the practical reliability of AI grading in a realistic large-scale deployment setting. Accordingly, the original presentation emphasized interpretable metrics (MAE, agreement rates, independent review). The numbers appear in the quoted sentence from **6.1.2** refer to rounded MAE and within-1 rate ranges across quizzes, not pooled estimates.
>
> We agree that additional statistical summaries strengthen the analysis. In response, we now report **95% confidence intervals (CI, computed via bootstrap with 5,000 resamples)** and paired-comparison statistics.
>
> **95% CI for MAE:**
>
> | Course | Quiz  | n_pairs | MAE      | 95% CI_low   | 95% CI_high  |
> |--------|-------|--------:|---------:|---------:|---------:|
> | A      | Quiz2 |     540 | 0.715741 | 0.611111 | 0.828704 |
> | A      | Quiz3 |     265 | 0.711321 | 0.590566 | 0.839623 |
> | A      | Quiz4 |     308 | 0.782468 | 0.665584 | 0.905885 |
> | A      | Quiz5 |     255 | 0.645098 | 0.527451 | 0.770637 |
> | A      | Quiz6 |      88 | 1.062500 | 0.869318 | 1.272727 |
> | B      | Quiz2 |     402 | 0.708458 | 0.634695 | 0.786452 |
> | B      | Quiz3 |     618 | 0.470227 | 0.419417 | 0.522168 |
> | B      | Quiz4 |     453 | 0.828918 | 0.727373 | 0.935982 |
> | B      | Quiz5 |     624 | 0.688221 | 0.631330 | 0.746797 |
> | B      | Quiz6 |     392 | 0.802296 | 0.709184 | 0.899235 |
>
> **95% CI for within-1 agreement rate:**
>
> | Course | Quiz  | n_pairs | Within-1 (%) | 95% CI_low | 95% CI_high |
> |--------|-------|--------:|-------------:|-----------:|------------:|
> | A      | Quiz2 |     540 |       81.667 |     78.519 |      84.815 |
> | A      | Quiz3 |     265 |       82.264 |     77.358 |      86.793 |
> | A      | Quiz4 |     308 |       76.623 |     71.753 |      81.169 |
> | A      | Quiz5 |     255 |       82.745 |     78.039 |      87.059 |
> | A      | Quiz6 |      88 |       68.182 |     57.955 |      78.409 |
> | B      | Quiz2 |     402 |       81.343 |     77.363 |      85.075 |
> | B      | Quiz3 |     618 |       85.599 |     82.686 |      88.350 |
> | B      | Quiz4 |     453 |       71.965 |     67.770 |      76.159 |
> | B      | Quiz5 |     624 |       78.365 |     75        |      81.414 |
> | B      | Quiz6 |     392 |       77.041 |     72.449 |      81.122 |
>
> **95% CI for feedback accuracy:**
> Over 3,851 responses, proportions rated **Correct / Acceptable / Incorrect**  (with **95% CI**) are
> 79.79% [78.47, 81.02], 9.55% [8.65, 10.46], and 10.67% [9.71, 11.69].
>
> **Intersection-aligned subset (171 solutions).**
>
> **GPT-4.1-mini vs. Reviewer 1/2**:
> **MAE**: 0.4006/0.4814,
>
> **within-1**:  91.30%/90.68%.
>
> **Reviewer 1 vs. Reviewer 2**:
> **MAE**: 0.4845,
>
> **within-1**: 90.06%.
>
> Thus, **AI–human** disagreement matches **human–human** variability.
>
> **Paired t-tests (171 subset).**
>
> **GPT-4.1-mini vs. Reviewer 1:** mean diff −0.295, p = 4.45×10⁻¹⁰  (AI slightly stricter).
> **GPT-4.1-mini vs. Reviewer 2:** mean diff +0.1025, p = 0.0747 (not significant at 0.05).
> **Reviewer 1 vs. Reviewer 2:** mean diff −0.3975, p = 9.33×10⁻¹³.
>
> This aligns with the full dataset (~4000 responses):
> AI < TA: 39.70%, AI = TA: 48.44%, AI > TA: 11.86%, indicating a mild tendency toward stricter grading with small overall discrepancy.
>
> Overall, while small systematic differences may exist, their magnitude is limited and AI–human agreement remains comparable to human variability.
>
>
> **2. The max-rule selection**
>
> - The max-rule captures diverse valid solution paths: fixed rubric for canonical solutions, flexible rubric for non-standard correct reasoning.
>
> - We acknowledge it could in principle introduce upward bias. However, empirical results do not support this: AI grading is slightly more strict on average, and **AI–human** disagreement on the aligned subset is comparable to **human–human** disagreement.
>
> - Moreover, nearly 90% of AI scores are no higher than TA scores under the max-rule, indicating no systematic inflation. Simpler alternatives (e.g., averaging) would tend to lower scores further.
>
> - A broader comparison of aggregation rules (e.g.,  other monotone score transforms) would be interesting future work.
>
>
> **3. Exemplar selection for rubric generation**
>
> Exemplars refer to a small library of representative problems/rubrics grouped by standard single-variable calculus types (e.g., limits, derivatives, integrals). For a new problem, the closest exemplar, together with the problem statement and a reference solution, is used to prompt rubric generation. The draft is then lightly edited by a human before deployment.
>
> ---
> **Correction (for all reviewers)**
>
>  The survey data come from two sections of Course A (not A vs. B); this affects interpretation but not results.

---

> > ### Author Rebuttal · Reviewer_fRQp · 2026-04-04
> >
> > Thanks for the clarifications and the additional results. Considered together, these strengthen the study's overall soundness and its presentation.

---

### Official Review · Reviewer_bU29 · 2026-03-13

**Soundness:** 4
**Presentation:** 4
**Significance:** 4
**Originality:** 3
**Overall Recommendation:** 5
**Confidence:** 4

**Summary:**

The paper presents a large-scale, real-world deployment study of an AI-assisted grading system for undergraduate single-variable calculus. Operating across three academic terms with nearly 800 students and thousands of handwritten free-response quiz submissions, the system utilizes a two-stage pipeline: (1) an LLM-based Optical Character Recognition (OCR) module (using GPT-4.1-mini) to transcribe handwritten math into LaTeX, and (2) a rubric-guided LLM evaluator to assign partial credit and generate formative feedback.

Because grading complex, open-ended math problems lacks a single objective "ground truth," the authors evaluate the system from multiple perspectives: alignment with official Teaching Assistant (TA) scores, student surveys on feedback utility, and rigorous independent human reviews. The results indicate strong score alignment with TAs (mean absolute errors generally under 1.0 point) and high feedback acceptance rates (almost 90% of AI feedback rated "correct" or "acceptable" by independent reviewers). The study provides valuable insights into prompt engineering to mitigate OCR hallucination and autocorrection, proposes a "max-rule" dual-rubric system (flexible vs. fixed) to handle diverse solution strategies, and lays the groundwork for a future public benchmark.

**Compliance With Llm Reviewing Policy:**

Affirmed.

**Key Questions For Authors:**

n/a

**Limitations:**

yes

**Strengths And Weaknesses:**

**Soundness**: The authors correctly identify that grading open-ended math proofs lacks an absolute "ground truth." Therefore, their multi-perspective evaluation method (TA scores + Student Surveys + Independent Expert Review) is methodologically sound and highly appropriate for the problem domain. The paper is honest and thorough about its failures. The detailed examination of OCR failures (e.g., misinterpreting nested fractions) and prompt-induced errors (e.g., the LLM "autocorrecting" a student's arithmetic mistake before grading it) adds significant technical depth.

**Presentation**: The overall structure of the paper is logical. Figure 1 provides a clear, high-level overview of the pipeline (Region Segmentation -> OCR -> Grading). The authors provide concrete examples of student handwriting, OCR outputs, and prompt templates. Examples like the "autocorrection issue" (Example 4) clearly demonstrate why specific prompt instructions were necessary.

**Significance**: Most current AI education benchmarks focus on multiple-choice or short-answer questions. This paper tackles the much harder problem of multi-step, handwritten partial-credit grading, pushing the frontier of applied LLM capabilities. The authors' promise to release the dataset (in both "clean" and "noisy" tracks) is a significant contribution. It will allow future researchers to test OCR and reasoning models on authentic, messy human data.

**Originality**: The observation that LLMs will implicitly "fix" a student's math error during the OCR transcription phase—thereby ruining the subsequent grading phase—is a highly original and insightful finding that future builders of similar systems must address.

However, there are some concerns that need to be considered:
- While the authors compare GPT-4.1-mini to Mathpix for the OCR step, they provide very little comparison for the grading step.

---

> ### Author Rebuttal · Authors · 2026-03-31
>
> **1. Regarding comparison for the grading step**
>
> We thank the reviewer for this insightful observation. Following this suggestion, we performed an additional **controlled comparison** to clarify the role of **OCR** in **downstream grading performance**.
>
> This provides a very useful way to quantify, to some extent, the downstream impact of OCR quality on grading accuracy.
>
> In our system, **Mathpix** is used only for OCR, while grading is performed by the **LLM (GPT-4.1-mini)**. Thus, a direct "**Mathpix vs. GPT-4.1-mini**" comparison is meaningful only at the OCR stage; for the grading stage, the relevant controlled comparison is to fix the grading model and vary the OCR input.
>
> **2. Effect of OCR quality on downstream grading**
>
> On the same **171-solution subset**, with the grading model fixed, replacing **GPT-4.1-mini OCR** with **Mathpix OCR** substantially worsens agreement with human reviewers:
>
>
> | Pair                             | MAE    | Mean Diff | \|d\| ≤ 0.5 (%) | \|d\| ≤ 1.0 (%) |
> |----------------------------------|--------|-----------|-----------------|-----------------|
> | Mathpix+4.1 vs Reviewer 1        | 0.6832 | -0.6211   | 61.49           | 77.64           |
> | Mathpix+4.1 vs Reviewer 2        | 0.7019 | -0.2236   | 60.87           | 79.50           |
> | GPT-4.1-mini vs Reviewer 1       | 0.4006 | -0.2950   | 80.75           | 91.30           |
> | GPT-4.1-mini vs Reviewer 2       | 0.4814 | +0.1025   | 76.40           | 90.68           |
> | o3-mini vs Reviewer 1            | 0.2422 | -0.1863   | 92.55           | 97.52           |
> | o3-mini vs Reviewer 2            | 0.4658 | +0.2112   | 72.05           | 93.17           |
> | Reviewer 1 vs Reviewer 2         | 0.4845 | -0.3975   | 74.53           | 90.06           |
>
>
> These results show that **OCR quality** alone has a substantial downstream effect on grading performance, as expected for such an important upstream component of the pipeline.
>
> We will clarify this point more explicitly in the revision.
>
> ---
>
> **Correction (for all reviewers)**
> The survey data come from two sections of Course A (not A vs. B); this affects interpretation but not results.

---

> > ### Author Rebuttal · Reviewer_bU29 · 2026-04-06
> >
> > Thanks for the additional experiments.

---

### Official Review · Reviewer_vxAJ · 2026-03-15

**Soundness:** 1
**Presentation:** 3
**Significance:** 3
**Originality:** 2
**Overall Recommendation:** 3
**Confidence:** 4

**Summary:**

The authors present results from a large scale empirical study of AI grading in single-variable calculus for a wide range of classes from a major U.S. public research university. They use OCR to convert the handwritten math into LaTeX and provide a structured rubric to an LLM for automatic generation of scores and feedback. GPT-4.1-mini is used as the standard model with o3-mini being used for more difficult cases. They then validate the generated scores and feedback across actual TA scores and human reviewers.

**Compliance With Llm Reviewing Policy:**

Affirmed.

**Final Justification:**

I'm very torn on this but I think I'll ultimately go up to at least a weak reject. As I previously said, there is alot of good stuff in this paper. The approaches and results from applying an automatic grading system like this in a real world setting is definitely valuable. But my big hesitation still remains, that the benchmark itself is still pending instead of something ready for release now.

**Key Questions For Authors:**

Have you tested whether the autocorrection and hallucination failure modes you document persist in more recent models?

What is the concrete timeline and release plan for the benchmark? Without the benchmark, how should we take the "toward a benchmark" framing?

Given the substantial inter-reviewer disagreement documented in the appendix, have you considered doing this on a AI-human level as well to see if the same disagreements hold?

**Limitations:**

Yes, but I'll point to the benchmark question as something to be addressed.

**Strengths And Weaknesses:**

Strengths and Weaknesses:

I’m going to be blunt here, I think there is alot of value in this paper but this just might not be the venue for it. I’ll provide an initial assessment of the main problems I see, but then also explain why I think it would make this paper perfect for a more education-focused venue such as Learning@Scale.

1.)  The biggest strength of the paper is that it has identified a very clear problem and provides explicit results from an actual in-the-field deployment. You might argue that OCR is unnecessary and the students should just be writing in latex anyway, but with the current issues around academic honesty and plagiarism, not to mention consideration for low-income environments where computer access may be more limited, there’s still immense value in this kind of system. (Significance +)

However from a purely machine learning perspective, there is minimal contributions here. Rubric-based scoring to improve LLM grading accuracy isn’t particularly novel. This is an extensive and very rigorous evaluation for methods whose effectiveness is already reasonably well-documented. (Originality −, Significance −)

That being said, this study is still a goldmine of information since there is a whole host of reasons, PII, inaccuracy in grading, fairness, etc, why LLMs aren’t as widely leaned on in the classroom. (Significance +)

2.) The paper exposes interesting failure modes, ie the autocorrection example. This is something very important for an educational setting but I think it would need to be explored if these same failure modes still occur in more recent LLMs. I don’t think it would need to be a substantial evaluation but at least a small handful of samples where it's tested if GPT 5+ makes the same error would go a long way. It’d be extremely insightful to see if this is an error that still holds. (Soundness −)

3.) I think it needs to be addressed but this is ultimately a benchmark paper without a benchmark. (Soundness −, Significance −)

4.) This is a finer point but the evaluation methodology is inherently chasing a very noisy target. There’s the obvious issue of students self-selecting for the survey. Appendix A shows that even the graders themselves tend to disagree with each other. An argument might be made that the distribution of high school through graduate-level students is too noisy a range but in actual practice, where this paper shines, that’s just the pool of people who would likely be grading the assignments. There’s a principled approach argument here where you should at least try to match the original distribution (human graders) before optimizing that I would make if the focus is on the algorithm. But if the focus is on the classroom, having a system that produces less disagreement is a positive in a setting where fairness is also a first-class optimization objective. (Soundness −, Significance +)

---

> ### Author Rebuttal · Authors · 2026-03-31
>
> We thank the reviewer for the thoughtful and constructive feedback. We address the main concerns below.
>
> **1. The paper’s contribution from an ML perspective**
>
> We agree that rubric-guided prompting itself is not novel in isolation. While the application is educational,  this work studies an important ML problem: evaluation of LLM systems under noisy OCR inputs, partial-credit outputs, and absence of ground truth. Our contribution   lies in **application-driven** machine learning at the **problem and system level**, namely:
>
> - a **large-scale, real-course deployment** of LLM-based grading on authentic handwritten college mathematics across multiple terms,
> - a **new evaluation setting** involving OCR-conditioned reasoning, partial-credit grading, feedback generation, and **no single ground-truth label**,
> - a **systematic empirical study of failure modes and design principles**, including autocorrection, rubric interaction, and OCR-induced bias, and
> - an evaluation that goes beyond score agreement to also assess **feedback quality**, which is essential in practice and substantially harder than numerical grading alone.
>
> We believe this is relevant to ICML because evaluation, reliability, and human-aligned performance under realistic constraints are important machine learning questions, especially in high-impact applications. We will clarify and emphasize in the revision that the main contribution is **establishing and characterizing this domain**, rather than proposing a new algorithm.
>
> **2. OCR failure modes and newer models.**
>
> We thank the reviewer for this suggestion.
>
> We conducted limited exploratory checks with **GPT-5** when it first became available last August. In our preliminary tests, **GPT-5** did not outperform **GPT-4.1-mini** overall on this **OCR** task, so we did not adopt it in the main pipeline.
>
> We conducted additional checks as suggested.
>
> - **Autocorrection:** persists in newer models. Using **GPT-5.1** on the Appendix A.5 example, the unwanted correction (3+2=5) still occurs in repeated runs.
> - **Blank hallucination:** improved in newer models. With an explicit prompt instruction (e.g.,“if the image is blank, then output blank”), **GPT-5.1** is stable, while **GPT-4.1-mini** remains less consistent under the same setting.
>
> These results suggest that some failure modes persist while others improve. Since the mitigations used in our paper are simple and require little additional cost or implementation effort, we retain them across models.
>
> **3. Benchmark-related question.**
>
> We appreciate this point and agree that the wording could be clearer. Our intent was to indicate that this work establishes the foundation for a benchmark, including dataset construction, evaluation protocol, and labeling pipeline.
>
> We are preparing a preliminary public release for summer, organized by representative problem types, with 20–30 samples per category. Most core components of the benchmark are already constructed and used internally; the remaining steps before release are
>
> **(1)** privacy/compliance review for handwritten data and
>
> **(2)** an additional round of expert label verification.
>
> **4. AI–human vs human–human agreement.**
>
> Following the suggestion, we evaluated this on the **171-solution subset** with two independent reviewers.
>
> | Pair                             | MAE    | Mean Diff | \|d\| ≤ 0.5 (%) | \|d\| ≤ 1.0 (%) |
> |----------------------------------|--------|-----------|-----------------|-----------------|
> | GPT-4.1-mini vs Reviewer 1       | 0.4006 | -0.2950   | 80.75           | 91.30           |
> | GPT-4.1-mini vs Reviewer 2       | 0.4814 | +0.1025   | 76.40           | 90.68           |
> | o3-mini vs Reviewer 1            | 0.2422 | -0.1863   | 92.55           | 97.52           |
> | o3-mini vs Reviewer 2            | 0.4658 | +0.2112   | 72.05           | 93.17           |
> | Reviewer 1 vs Reviewer 2         | 0.4845 | -0.3975   | 74.53           | 90.06           |
>
> Thus, **AI–human** agreement is **comparable** to **human–human** agreement on this controlled subset. We will clarify this in the revision.
>
> **5. Noisy targets and classroom realism.**
>
> We agree with the reviewer’s point. Indeed, one of the central challenges of this project is that grading lacks an absolute ground truth, and some variability is inevitable even among mathematically qualified human reviewers.
> Our goal is therefore not to recover a single “correct” label, but to develop a grading system that is reliable in practice.
> In this setting, comparison with human reviewers remains an important operational reference for deployment, even if it should not be interpreted as a perfect gold standard.
>  We will clarify this perspective in the revision.
>
> ---
>
> **Correction (for all reviewers)**
> The survey data come from two sections of Course A (not A vs. B); this affects interpretation but not results.

---

> > ### Author Rebuttal · Reviewer_vxAJ · 2026-04-02
> >
> > This is an initial acknowledgement as I feel like I need to sit with this a bit more.
> >
> > My concerns about algorithmic impact still remain as well as 'benchmark without a benchmark'. I understand the procedures that would be required for the release of student data and do respect the authors doing their due diligence with regards to this. However, this also raises the question of "Why submit when a major component of the work is still in production?"
> >
> > I do appreciate the additional experiments and do acknowledge the value of this work. Again, I do have concerns about the misalignment of venue but am down weighting this in terms of how much I care about it for my final score.
> >
> > The points made by reviewer 4h6W are additional hesitations for me. Ultimately, due to the nature of the setting, reproducibility is very difficult. The systematic bias also seems like a major concern. Again, 'benchmark without a benchmark' issue, which is the biggest hurdle I'm having trouble with in terms of adjusting my score upwards.
> >
> > I'm leaning towards raising my score to at least a weak reject but I think I've made my biggest hesitation clear here.
> >
> > Edit: Softened some language.

---

### Decision · Program_Chairs · 2026-04-30

**Decision:**

Accept (regular)

**Comment:**

This paper presents a large-scale (800 students in three academic terms), real-world study of AI-assisted grading systems for undergraduate single-variable calculus. The pipeline converts students' handwritten math into LaTeX and provides a structured rubric to an LLM for automatic generation of scores and feedback. AI's scores and feedback show high consistency with actual TA scores and human reviewers. The study also discussed a few directions for improvement and outlined a benchmark for future work.

The paper focuses on an important problem for AI-TA alignment in education. Most reviewers acknowledged the novelty of the in-the-field study with rare real-world data. They especially appreciate the analysis of OCR failure modes, the engineering insights, and the comprehensive evaluation perspectives. However, they also pointed out that no benchmark has been released with the paper, and the benchmark is still a proposal. There is a potential risk of systematic bias and demographic fairness. A few reviewers also raised concerns about the max-rule evaluation for aggregation. There were some details that needed to be clarified, and the rebuttal successfully addressed them. Based on the final justifications, most reviewers are satisfied with the author responses, but a few main concerns remain (e.g., the lack of a benchmark).

Considering the discussion and final justifications, I think the paper's main contribution is important and original, though there are a few places that need to be improved. I recommend it for acceptance and suggest that the authors carefully polish the paper and address the concerns raised in reviews.